# Automated Black-box Prompt Engineering for Personalized Text-to-Image Generation

## Abstract

Prompt engineering is an effective but labor-intensive way to control text-to-image (T2I) generative models. Its time-intensive nature and complexity have spurred the development of algorithms for automated prompt generation. However, these methods often struggle with transferability across T2I models, require white-box access to the underlying model, or produce non-intuitive prompts. In this work, we introduce PRISM, an algorithm that automatically produces human-interpretable and transferable prompts that can effectively generate desired concepts given only black-box access to T2I models. Inspired by large language model (LLM) jail-breaking, PRISM leverages the in-context learning ability of LLMs to iteratively refine the candidate prompt distribution built upon the reference images. Our experiments demonstrate the versatility and effectiveness of PRISM in generating accurate prompts for objects, styles, and images across multiple T2I models, including Stable Diffusion, DALL-E, and Midjourney.

## 1 Introduction

An important goal of generative modeling is to design algorithms capable of steering generative models to produce desired output images. Early attempts, which often centered on particular architectures or tasks, were largely characterized by manually-curated data collection, fine-tuning, or retraining from scratch (Srivastava & Salakhutdinov, 2012; Mirza & Osindero, 2014; Isola et al., 2017; Zhu et al., 2017). These requirements are often costly, and the resulting solutions usually do not transfer well between models. Thus despite the promise of these methods, efficient and generalized algorithms for controllable generation remain sought after.

Today, perhaps the most popular approach for controllable generation is to guide the generation process with a piece of textual information, or prompt, that describes the properties of the desired output using text-to-image (T2I) generative models (Rombach et al., 2022; Yu et al., 2022). Through text, T2I models allow users to quickly and easily describe a wide variety of concepts, and users can more efficiently explore the behavior of their model through a myriad of strategies Chao et al. (2023); Wen et al. (2023). The predominant method for obtaining such input text is to manually design candidate prompts in an iterative, trial-and-error fashion, a process known as *prompt engineering*, based on what the user (prompt engineer) *believes* will lead to a desirable output. Unfortunately, these practices are often sensitive to different phrasings (Webson & Pavlick, 2022), require expert domain knowledge, and are notably inefficient as they necessitate a human in the loop.

Motivated by the drawbacks of manual prompt engineering, a recent line of work known as *personalized* or *subject-driven* T2I generation has sought to automate the controllable generation pipeline. Given a collection of reference images that capture specific concepts, such as artistic style or shared objects, personalized T2I algorithms are designed to produce images that reflect those concepts illustrated in the reference images. While personalized T2I methods often involve fine-tuning or retraining the underlying T2I model (Ruiz et al., 2023; Chen et al., 2023; Shi et al., 2023), several approaches focus specifically on automating prompt engineering to generate effective prompts. Unfortunately, existing algorithms in this spirit tend to require pre-collected, architecture-specific keywords[1] or white-box, embedding-based optimization (Gal et al., 2023; Mahajan et al., 2023), leading to non-interpretable prompts (Wen et al., 2023) and preclude the possibility of directly generating prompts for closed-source T2I models (e.g., Midjourney or DALL-E).

---

[1] https://github.com/pharmapsychotic/clip-interrogator

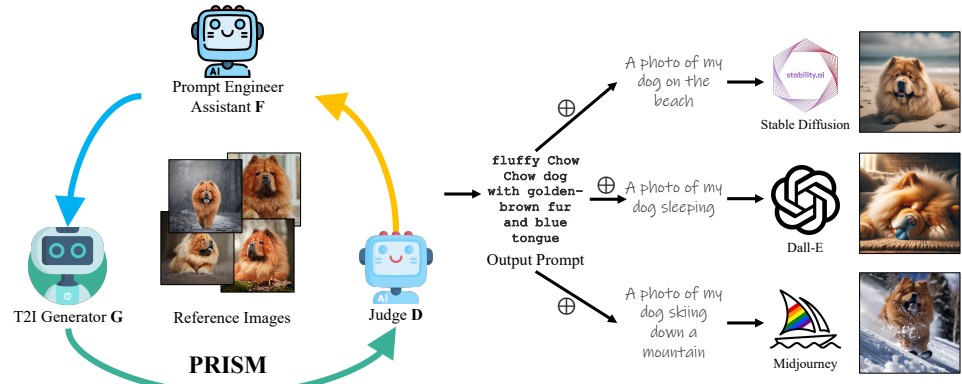

Figure 1: Given a set of reference images, our method, PRISM, is capable of creating human-interpretable and accurate prompts for the desired concept that are also transferable to both open-sourced and closed-sourced text-to-image models. ⊕ denotes prompt concatenation.

In order to address these shortcomings, we propose ***Prompt Refinement and Iterative Sampling Mechanism*** (PRISM), a new automated prompt engineering algorithm for personalized T2I generation. A key observation is that prompt engineers repeat the process of updating their "belief" of what makes an effective prompt based on the difference between their desired results and the generated images from previous iterations. Inspired by jailbreaking attacks on large language models (LLMs) (Chao et al., 2023) and LLMs as optimizers (Pryzant et al., 2023), we design an algorithm that operates with only limited human input, is capable of generating human interpretable and editable prompts, makes minimal assumptions about the underlying T2I model, and generalizes across different T2I models, including popular black-box models such as DALL-E and Midjourney.

Given a set of reference images, our method first generates an initial prompt and its corresponding image using a vision-language model (VLM) as "prompt engineer assistant" and a T2I generator. We then obtain a score indicating the visual similarity of the generated image and the reference image with respect to the targeting concept via another VLM as judge. Leveraging LLMs' in-context learning abilities (Shin et al., 2020; Zhou et al., 2023; Yang et al., 2024), we instruct the prompt engineer assistant VLM to update the candidate prompt distribution based on the previously generated prompt, images, and the evaluation scores. This processing, shown in Figure 1, is then repeated for a predetermined number of iterations. In the end, PRISM outputs the best-performing prompt by re-evaluating the top prompts generated from this process. In this way, PRISM seamlessly integrates iterative reasoning into the image generation process, much like a real prompt engineer. Our approach can therefore go beyond basic image-to-image transformations and conventional single-shot methods, providing a more versatile and robust framework for generating images that are both visually precise and contextually relevant.

Experimentally, our method shows significantly better generalizability and transferability as we achieve the best performance in almost all metrics when experimenting with closed-source models in comparison to baselines including Textual Inversion (Gal et al., 2023), PEZ (Wen et al., 2023), BLIP2 (Li et al., 2023) and CLIP-Interrogator[1]. Our results also indicate that PRISM consistently outperforms existing methods with respect to human-interpretability while maintaining high visual accuracy. Finally, we demonstrate that the strong human interpretability makes the prompts generated by PRISM easily editable, unlocking a wide array of creative possibilities in real life.

## 2 RELATED WORKS

**Controllable T2I generation** Several methods tackle conditional image generation in a training-free manner by using pretrained diffusion models as priors (Meng et al., 2022; Chung et al., 2023; Song et al., 2022; He et al., 2023), and analogous approaches exist for T2I diffusion models (Yu et al., 2023; Rout et al., 2023; He et al., 2024). However, these methods assume that the controllability objectives can be formulated as differentiable loss functions, require access to model parameters and involve complex hyperparameter tuning. Another class of approaches (Zhang et al., 2023; Ye et al., 2023; Ruiz et al., 2023; Chen et al., 2023; Shi et al., 2023) also improve the controllability of pretrained T2I models, but they require expensive fine-tuning or re-training of the underlying model

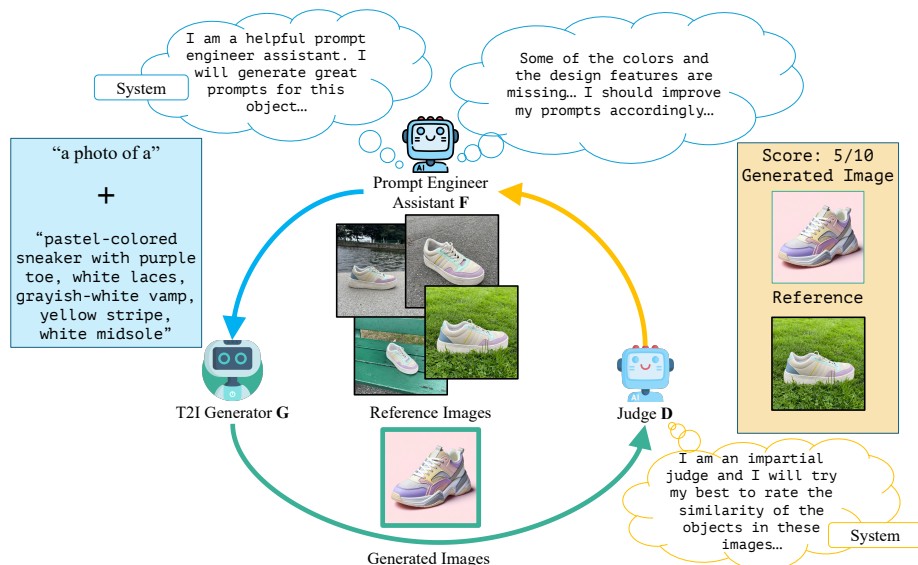

Figure 2: An illustration of PRISM. "System" indicates the system prompt setups for the VLMs.

every time they are applied to a new task. Prompt tuning methods (Gal et al., 2023; Wen et al., 2023; Mahajan et al., 2023) are in the same spirit as this paper, as they do not require training of the T2I model and condition generations on given reference images. However, unlike PRISM, these methods require access to the underlying model parameters or produce non-interpretable prompts.

**Prompt engineering**   Manual prompt engineering is a popular approach to eliciting desired behaviors from large pre-trained models because it uses little or no data and does not require fine-tuning (Radford et al., 2021; Brown et al., 2020). However, major drawbacks of manual prompt engineering include its laborious nature, its reliance on domain expertise, and its sensitivity to phrasings (Lu et al., 2022; Webson & Pavlick, 2022). To address this issue, several methods have been proposed to construct prompts in an automated manner (Shin et al., 2020; Gao et al., 2021; Zhou et al., 2022; 2023; Manikandan et al., 2023; Pryzant et al., 2023; Yang et al., 2024), and some have applied similar techniques to various downstream tasks (Mañas et al., 2024; Liu et al., 2024; Yang et al., 2023; Hao et al., 2024). In particular, Liu et al. (2024) applied the algorithm they designed for image classification to image inversion. Moreoever, LLM jailbreaking focuses on automatically designing prompts that elicits specific content (often objectionable or illicit) from a targeted LLM (Zou et al., 2023; Wei et al., 2024; Robey et al., 2023; Liu et al., 2023). A particularly relevant work is Chao et al. (2023), which uses an auxiliary LLM to iteratively construct jailbreak prompts. Our method builds on this idea to create prompts to generate images satisfying the desired criteria.

## 3   METHOD

### 3.1   PROBLEM STATEMENT

First, let $x \in \mathcal{X}$ denote an image, and $y \in \mathcal{Y}$ denote a textual prompt. Given a collection of reference images $\{x_i\}_{i=1}^{M}$, a prompt engineer $\mathbf{F} : \mathcal{X} \to \mathcal{Y}$ samples a candidate prompt $y$ corresponding to each reference image $x$, i.e., $y \sim p_{\theta_{\mathbf{F}}}(y \mid x)$. A T2I generative model $\mathbf{G} : \mathcal{Y} \to \mathcal{X}$ then uses this candidate prompt to generate a new image, $x \sim p_{\theta_{\mathbf{G}}}(x \mid y)$, and a judge model $\mathbf{D} : \mathcal{X} \times \mathcal{X} \to [0, 1]$ then scores the visual similarity between the images based on some criteria. Our goal is then to find the best prompt:

$$y^{\star}\left(\{x_i\}_{i=1}^{M}\right) = \arg\max_{y \in \mathcal{Y}} \sum_{i=1}^{M} s(x_i, y), \tag{1}$$

where $s(x_{\text{target}}, y) = \mathbb{E}_{x \sim p_{\theta_{\mathbf{G}}}(x|y)} \left[ \mathbf{D}(x, x_{\text{target}}) \right]$.

The criteria can be any visual similarity metric that may or may not be easy to specify in a closed form, including "*how similar are the main objects in the images*" or "*how similar are the styles of the image*" or "*how similar are the two images in general*". The resulting $y^{\star}$ should be able to generate

---

**Algorithm 1** Prompt Refinement and Iterative Sampling Mechanism (PRISM)

---

1: **Input:** $N$ streams, $K$ iterations, $\{x_i\}_{i=1}^{M}$ reference images
2: **Output:** Best prompt $y^\star$ based on total score
3: **for** $n = 1$ **to** $N$ **in parallel do**
4:     **for** $k = 1$ **to** $K$ **do**
5:         Randomly sample an $x_{k,n}$ from $\{x_i\}_{i=1}^{M}$
6:         **F** samples $y_{k,n} \sim p_{\theta_{\mathbf{F}}}(y \mid x_{k,n})$
7:         **G** samples $\hat{x}_{k,n} \sim p_{\theta_{\mathbf{G}}}(x \mid y_{k,n})$
8:         **D** calculates an in-iteration score $s'(x_{k,n}, y_{k,n}) = \mathbf{D}(x_{k,n}, \hat{x}_{k,n})$
9:         Update $p_{\theta_{\mathbf{F}}}$ based on $x_{k,n}, \hat{x}_{k,n}, y_{k,n}, s'(x_{k,n}, y_{k,n})$ and the chat history of stream $n$
10:     **end for**
11: **end for**
12: Collect the subset $\{y_c\}_{c=1}^{C}$ with the $C$-best in-iteration scores
13: Re-evaluate this subset with total score $\sum_{i=1}^{M} s(x_i, y_c)$
14: Return the prompt with the best total score. In case of a tie, return the prompt with the highest log likelihood.

---

an image that is very close to the reference images based on the criteria with some (possibly unseen) T2I models $p_\theta(x \mid y)$.

## 3.2 Algorithm

Our method, Prompt Refinement and Iterative Sampling Mechanism (PRISM), is an iterative process that repeats a prompt refinement subroutine for $K$ iterations in $N$ parallel streams, where $N \times K$ is a predetermined compute budget. At iteration $k$, the $n$-th stream of PRISM randomly selects a reference image $x_{k,n}$ from $\{x_i\}_{i=1}^{M}$ and uses **F** to sample a candidate prompt $y_{k,n}$ from $p_{\theta_{\mathbf{F}}}(y \mid x_{k,n})$. Then it queries **G** to generate a single $\hat{x}_{k,n}$ from $y_{k,n}$ with $p_{\theta_{\mathbf{G}}}(x \mid y_{k,n})$ and evaluate the prompt with **D** to obtain an in-iteration score $s'(x_{k,n}, y_{k,n}) = \mathbf{D}(x_{k,n}, \hat{x}_{k,n})$. At the end of the iteration, we use the generated $y_{k,n}$ and its score to update $p_{\theta_{\mathbf{F}}}(y \mid x)$. After the entire process, we collect the subset of $\{y_c\}_{c=1}^{C}$ generated throughout this process that has the $C$-best in-iteration scores. Then we re-evaluate this subset with the total score $\sum_{i=1}^{M} s(x_i, y_c)$ and return the prompt with the best total score. If there is a tie, then we return the prompt with the highest log likelihood. The pseudocode and an illustration for this algorithm are outlined in Algorithm 1 and Figure 2 respectively.

The key difference between PRISM and prior methods is that PRISM updates the entire sampling distribution of prompts, whereas prior works (Gal et al., 2023; Wen et al., 2023; Mahajan et al., 2023) directly update the tokens of a single prompt or the embeddings of the prompt. We believe that maintaining the whole prompt distribution is beneficial as text-to-image generation is not a one-to-one operation, i.e. an image can be described by multiple different text prompts and the same text prompt can correspond to multiple differently generated images. Having access to the whole distribution allows the method to sample a more diverse range of prompts without starting from scratch and may also help the optimization escape potential local optima.

Since PRISM only requires samples from **G**, one may use any T2I model of their choice. However, careful consideration is needed when designing **F** and **D**, which we will elaborate on below.

## 3.3 Designing and updating **F** and $p_{\theta_{\mathbf{F}}}$

**What is $p(y \mid x)$?** In general, it is not obvious what the joint or the conditional distribution of all text and images is, so some form of approximation is unavoidable. In the context of image generation, a natural choice of the image-conditioned text distribution is an image captioning model. Traditional captioning models, however, fall short in controlled image generation for two primary reasons: **(1)** The level of detail necessary for generating specific images far exceeds what generic captioning models provide (Liang et al., 2023); **(2)** effective prompts for T2I models are often not grammatically correct sentences but rather collection of phrases that describe the details about the image, which generic captioning models are not trained to generate. For example, in Figure 5, the second reference image is generated by the prompt *"A broken robot lays on the ground with plants growing over it, somber, HD, hyper realistic, intricate detail"* with Stable Diffusion, but a caption

for this image will not include components like "HD" or "hyper realistic". As a result, instead of "a good description of an image", we wish to directly model "possible prompts that are used to generate this image".

**Desiderata** A desirable $\mathbf{F}$ can sample from a distribution $p_{\theta_{\mathbf{F}}}(y \mid x)$ that models "the prompt that can be used to generate this image", and it should also be easily updated if the current generation is suboptimal. Ideally, such an update can be done without any retraining or fine-tuning since these operations are generally expensive and incompatible with black-box T2I models.

**Vision-Language Models as $\mathbf{F}$** VLMs stand out as the ideal choice for $\mathbf{F}$ due to their ability to directly tailor the generation of prompts via system prompts and to adapt through in-context learning without requiring access to the model's parameters. Specifically, since the model can ingest both images and texts, we can incorporate the history of reference images, intermediate prompts, generated images, and the evaluation scores all in the context of the LLM. Then, the model can be prompted to jointly reason over all available information and perform in-context learning. The in-context learning facilitates iterative refinement of the prompt to update the posterior distribution based on feedback or even additional human instructions, without the need for model retraining. Concretely, the model would process how the image generative model is affected by different prompts, propose improvements, and create new prompts, much like a prompt engineer. This way, we can naturally incorporate iterative reasoning into the image generation process and go beyond simple image-to-image transformations and traditional single-shot generation, and thus offers a more robust and versatile framework for producing accurate and contextually relevant images. In practice, we design system prompts that explicitly condition the LLM to generate improvements and new prompts given the results from the previous iterations, similar to the chain-of-thought (Wei et al., 2022) and textual gradients (Pryzant et al., 2023) technique.

### 3.4 Designing the judge model $\mathbf{D}$

We have a wider range of choices for the judge model as long as it provides a notion of similarity between a pair of images. A simple solution is to use pre-trained discriminative models such as CLIP (Radford et al., 2021), and measure the distance in their embedding spaces. While these models have seen various degrees of success, they also come with inherent limitations – the discriminative objective (e.g., contrastive loss) does not incentivize the model to attend to fine-grained details, an issue similar to the shortcomings of using captioning models to generate prompts (Liang et al., 2023). Moreover, in image generation, the criteria of success can be nuanced and difficult to quantify through traditional similarity metrics yet can be effortlessly described in human language. Lastly, the similarity we wish to measure may only involve some part of the visual features (e.g. color), and not all applications share the same notion of similarity. If we want to use pretrained discriminative models, then we need to find a different model for each task, which can be inefficient.

In light of these challenges, an ideal judge model should be flexible for different kinds of criteria and can perform fine-grained analysis of the images. Once again, a VLM emerges as the perfect candidate: using system prompts and in-context learning, we can easily specify metrics that may be otherwise difficult to describe or evaluate and even intervene in the reasoning chain if we want to, and, more importantly, the same model can be applied to a wide range of tasks.

## 4 Experiments

### 4.1 Experimental Settings

**Implementation Details** For all of our experiments, we choose GPT-4V (OpenAI, 2023) as both the prompt engineer assistant model $\mathbf{F}$ and the judge $\mathbf{D}$. We also fix the T2I generator as SDXL-Turbo (Sauer et al., 2023) for all of our experiments. We design different system prompts for both $\mathbf{F}$ and $\mathbf{D}$ for each task and we provide details about the system prompts in the appendix.

We evaluate the prompts generated from PRISM and baselines with five different T2I models. In particular, we choose two open-sourced models, Stable Diffusion 2.1 (SD 2.1) and SDXL-Turbo, and two closed-sourced models, Dall-E 2 and Dall-E 3, to quantitatively measure the performance. We also qualitatively showcase results from Midjourney, which is another closed-sourced T2I platform. For SD 2.1 and SDXL-Turbo, we clip all prompt lengths to 77 due to their context length constraint.

We compare PRISM and baselines in two settings: personalized T2I generation and direct image inversion, and we will elaborate on the task definitions in their corresponding sections below. For

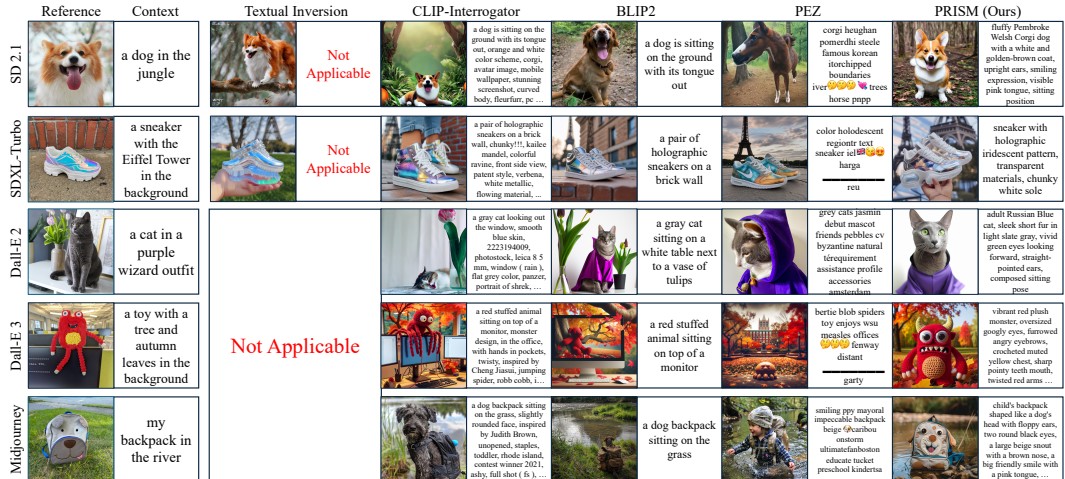

Figure 3: Qualitative results for personalized T2I generation on DreamBooth dataset.

personalized T2I generation, we use a maximum budget of 40 and report the quantitative results with $N = 10, K = 4$. For direct image inversion, we use a maximum budget of 30 and report the quantitative results with $N = 6, K = 5$.

**Baselines** We choose Textual Inversion (TI) (Gal et al., 2023), BLIP-2 (BLIP2) (Li et al., 2023), CLIP-Interrogator (CLIP-Int) and PEZ (Wen et al., 2023) as the baselines. Textual Inversion trains a "soft token" which cannot be directly translated into regular human language to represent the concepts in the reference images. BLIP-2 is the state-of-the-art image captioning model. CLIP-Interrogator[1] combines BLIP-2 captions with a suffix which is created by searching a pre-collected bank of keywords using CLIP (Radford et al., 2021) score. PEZ is a gradient-based optimization method that searches for the best combination of existing tokens in the vocabulary with CLIP similarity. For image inversion, we also include a VLM-based baseline, Liu et al. (2024), to demonstrate the effectiveness of our algorithm. Notice that TI requires training on individual models and CLIP-Int requires a pre-collected keyword bank, both of which provides unfair advantages over our setting.

**Evaluation Metrics** We evaluate the prompt interpretability using mean negative log-likelihood (NLL) calculated from Mistral 7B (Jiang et al., 2023). For image quality evaluation, we mainly measure the CLIP image similarity score (CLIP-I) to quantify the difference between the generated images and the reference images. Following Ruiz et al. (2023), we also use DINO V2 (Oquab et al., 2024) embedding similarity to calculate the object-sensitive image similarity for the personalized T2I generation task. We chose CLIP-ViT-L-14 and DINO-V2-Base as the base models. For Dall-E 2 and Dall-E 3, we also compare the number of times each method fails to pass its black-box safeguard. More failures indicate a higher potential to produce unsafe prompts. For each prompt, we allow 5 attempts before counting it as a failure.

### 4.2 PERSONALIZED TEXT-TO-IMAGE GENERATION

We first demonstrate PRISM's ability to find human-interpretable and accurate prompts to describe certain objects or styles in the task of personalized T2I generation. Given a set of reference images that depict the same concept (such as objects and style), this task requires the T2I model to synthesize images in new contexts while maintaining the original concept.

**Datasets** We use DreamBooth dataset (Ruiz et al., 2023) to quantitatively compare the performance in personalized T2I generation. DreamBooth dataset contains 30 daily objects and each subject has 4-6 images. For each subject, we adopt the 25 prompt templates curated by DreamBooth to create varying contexts and scenarios to test the fidelity of the subject representation in diverse settings. We generate 4 images for each subject and template combination with open sourced T2I models, and 1 image for each combination with closed sourced T2I models. For Textual Inversion, we follow the original setting to fill the templates and for all the other methods, we use the class noun to fill the template and the output prompts that describe these concepts serve as suffixes.

Table 1: Personalized T2I results on DreamBooth dataset. Bold fonts indicate the best score and underlines indicate the second best score.

| Method | Prompt | SD 2.1 | | SDXL Turbo | | Dall-E 2 | | | Dall-E 3 | | |
|---|---|---|---|---|---|---|---|---|---|---|---|
| | NLL ↓ | CLIP-I ↑ | DINO ↑ | CLIP-I ↑ | DINO ↑ | CLIP-I ↑ | DINO ↑ | Failed ↓ | CLIP-I ↑ | DINO ↑ | Failed ↓ |
| TI (SD 2.1) | - | 0.707 | 0.443 | - | - | - | - | - | - | - | - |
| TI (SDXL) | - | - | - | **0.771** | **0.504** | - | - | - | - | - | - |
| CLIP-Int | 4.361 | 0.733 | 0.446 | 0.756 | 0.490 | 0.711 | 0.464 | 13.3% | 0.619 | 0.386 | 1.1% |
| BLIP2 | 4.378 | 0.706 | 0.408 | 0.729 | 0.456 | 0.707 | 0.430 | **6.9%** | 0.655 | 0.377 | 0.3% |
| PEZ | 6.188 | 0.709 | 0.384 | 0.722 | 0.418 | 0.676 | 0.389 | 16.7% | 0.618 | 0.344 | 1.1% |
| PRISM (Ours) | **3.466** | **0.743** | **0.464** | 0.770 | 0.499 | **0.734** | **0.482** | **6.9%** | **0.734** | **0.464** | **0.1%** |

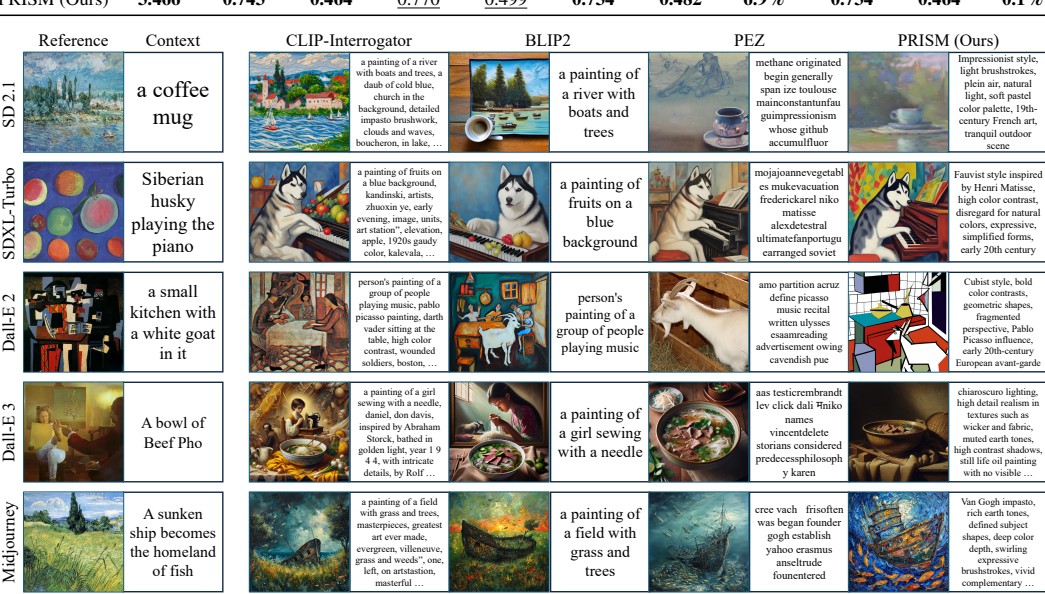

Figure 4: Qualitative results for personalized style T2I generation on Wikiart dataset.

We also qualitatively demonstrate the ability to represent a certain artistic style using Wikiart dataset (Tan et al., 2019). We use three images from each artist as reference images. To create diverse scenes, we follow He et al. (2024) and use descriptive prompts from PartiPrompts (Yu et al., 2022) as prefixes to the output prompts similar to the previous setting.

**DreamBooth Dataset Results** Table 1 and Figure 3 respectively show the quantitative and qualitative results on the DreamBooth dataset. As we can observe, PRISM achieves the best performance across the board except for the image similarity metrics for SDXL-Turbo.

In terms of object fidelity, we find PRISM to constantly achieve accurate depiction of the target subject while the baselines sometimes struggle to capture all fine-grained details like the colors of the animals and the shape of the shoe sole. And out of the four training-free methods we experiment with, PRISM is the only one that can attempt to tackle complicated objects such as the red monster toy and the dog-shaped backpack when all the other methods fail to generate even remotely similar objects. Due to the nature of their methodologies, BLIP-2 and CLIP-Interrogator also capture the background and other irrelevant elements in the scene when describing the objects. However, unlike our method, where we can directly specify the tasks and the judging criteria in the system prompts of the VLMs, there is no simple way to automatically filter out those irrelevant elements in BLIP-2 and CLIP-Interrogator's outputs. Even though Textual Inversion obtains marginally higher CLIP-I scores and DINO scores with SDXL-Turbo, it requires a lot more modeling assumptions than our method, and the new embeddings it learns are not transferable – not even to SD 2.1.

PRISM is the only method in our experiments that can produce fully human-readable prompts while providing enough relevant details. In particular, we can observe that PEZ renders completely indecipherable texts, BLIP-2 only describes the general scene but fails to mention any visual details and textual inversion is entirely not interpretable since it produces soft embeddings. Since CLIP-Interrogator combines the results from BLIP-2 and a CLIP search, it improves the interpretability

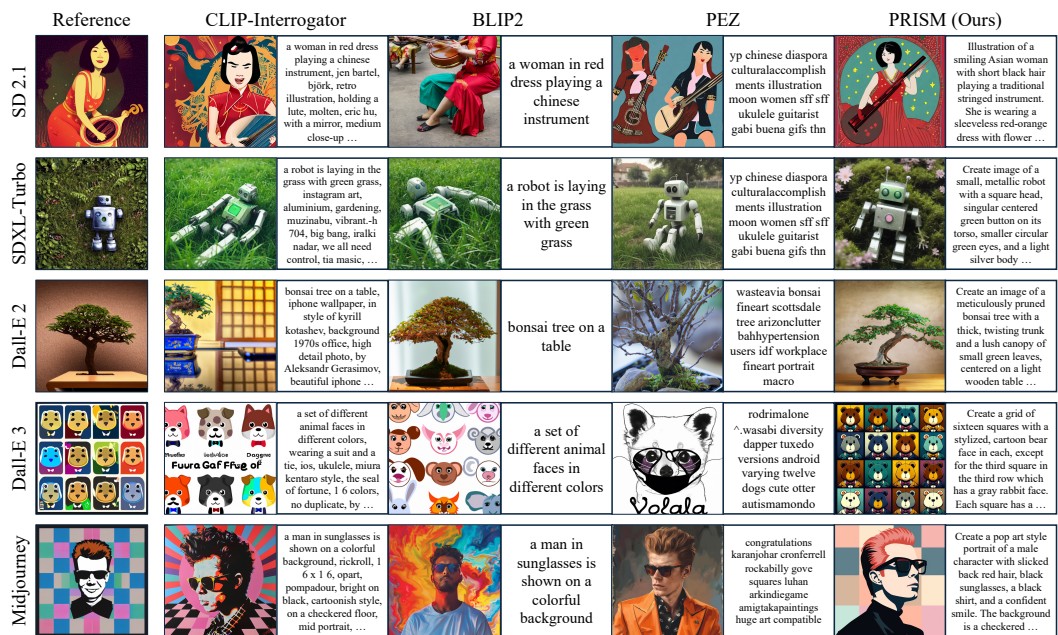

Figure 5: Image inversion results for different methods on different T2I models.

Table 2: Metrics for the image inversion results. old fonts indicate the best score and underlines indicate the second best score.

| Method | Prompt | SD 2.1 | SDXL TUrbo | Dall-E 2 | | Dall-E 3 | |
|---|---|---|---|---|---|---|---|
| | NLL ↓ | CLIP-I ↑ | CLIP-I ↑ | CLIP-I ↑ | Failed ↓ | CLIP-I ↑ | Failed ↓ |
| CLIP-Int | 4.193 | **0.800** | **0.783** | **0.761** | 17.0% | 0.719 | **0.0%** |
| BLIP2 | 4.299 | 0.710 | 0.707 | 0.687 | 2.0% | 0.695 | **0.0%** |
| PEZ | 6.736 | 0.746 | 0.726 | 0.616 | 3.0% | 0.635 | **0.0%** |
| Liu et al. (2024) | **2.520** | 0.713 | 0.720 | 0.689 | 0.0% | 0.732 | **0.0%** |
| PRISM (Ours) | 2.762 | 0.749 | 0.776 | 0.741 | 2.0% | **0.767** | **0.0%** |

over PEZ-like gradient search-only method. However, it still falls short in terms of human readability in comparison to our method.

When transferring the output prompts to black-box T2I models, our method shows even larger advantages over the baselines. We also observe that our method produces the fewest unsafe prompts judged by Dall-E safeguards, while the baselines fail to pass the safeguard up to 16.7% of the time.

**Wikiart Results** In Figure 4, we show a qualitative comparison between our method and baselines on the Wikiart dataset. We find that our method is capable of precisely identifying the genres, eras, and sometimes even the names of the artists when describing the style of the reference artworks. On the other hand, the baselines fail to recognize these crucial keywords, even when they have access to a pre-collected bank of words that is supposed to provide accurate descriptions of art styles. In addition, PRISM can provide other fine-grained details such as pen strokes style and color palettes in a human-interpretable way to better assist the generation of the target style.

## 4.3 DIRECT IMAGE INVERSION

To demonstrate the versatility of our method, we also compare PRISM the baselines in the task of direct image inversion. In this task, the goal is to directly find the prompt that can exactly generate the input image. Here the number of reference images is $M = 1$ and we aim to capture all aspects of the image, including the subjects, background, theme, style, and other details in the scene.

Table 3: Comparison with GPT-4V in both personalized T2I generation and direct image inversion experiments.

| Method | Image | | Object | |
|---|---|---|---|---|
| | NLL | CLIP-I | NLL | CLIP-I |
| GPT-4V | **2.356** | 0.756 | **3.393** | 0.757 |
| PRISM | 2.762 | **0.776** | 3.466 | **0.770** |

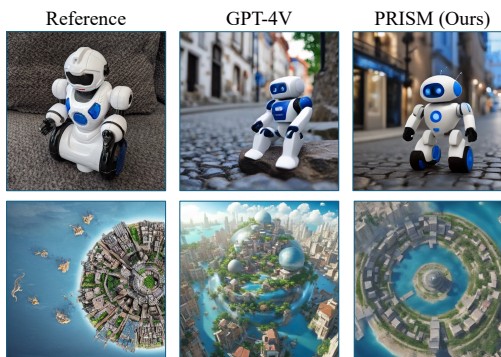

Figure 6: Qualitative comparison with GPT-4V.

**Datasets**   We use images from the DiffusionDB dataset (Wang et al., 2022) for the direct image inversion task. This dataset includes a wide variety of image pairs generated by Stable Diffusion and we choose a random sample of 100 images from the `large_random_10k` split on Huggingface.

**Results**   As shown in Table 2, we immediately see a significant improvement in the human-interpretability of inverted prompts using PRISM. While expected for methods, such as PEZ, which has no language prior, we also find that our method finds text that more closely aligns with a learned distribution of English language text (i.e. lower NLL) than CLIP-Interrogator and BLIP2.

When comparing the image quality, we first note that because all images in DiffusionDB are generated by Stable Diffusion, which is exactly the model design space of CLIP-Interrogator and PEZ, it gives significant modeling assumption advantages to these baselines over our method when testing on Stable Diffusion models. This advantage enables relatively high performance for these baselines on Stable Diffusion models, but it does not transfer well into other closed-sourced models. In fact, we can even observe that CLIP-Interrogator generates the highest quality images with SD 2.1, which is the weakest model in this comparison, and the lowest quality images with Dall-E 3, which is the strongest T2I model in this table. This phenomenon indicates that the design choices of CLIP-Interrogate and PEZ are heavily overfitted on Stable Diffusion, and have poor generalizability to other models. On the other hand, the prompts produced by PRISM generalize significantly better than the baselines and we achieve better results with more powerful T2I models. When compared with the other VLM baseline Liu et al. (2024), with which a thorough comparison is included in Section F, PRISM performs it in almost all metrics, indicating a superior algorithmic design.

Qualitatively, our method also provides prompts that are both semantically aligned with and can generate images that are visually similar to the reference. In particular, Figure 5, shows that we can find text that aligns with the image, even when those images have particularly unique features. For example, in Figure 5 Dall-E3 generated a grid of images of animal faces. Not only does the PRISM's prompt explicitly include a request for this grid structure, unlike our comparison methods, but it also takes into account the coloration of the background in the reference. In the second row of Figure 5, our method is also the only method that captures the small flowers in the grass, showcasing the capability of identifying and reflecting small fine-grained details from the reference.

### 4.4 ABLATION STUDY

**Comparison with GPT-4V**   While we can use any VLM (which we demonstrate in Appendix C.2), it is nonetheless useful to understand what benefits PRISM adds to an already capable foundation model like GPT-4V. Therefore, we compare our method with GPT-4V's zero-shot performance with the same system prompts for both tasks on SDXL-Turbo. We can see in Table 3 that PRISM consistently outperforms GPT-4V's zero-shot performance, although the latter is already compelling. In Figure 6, we can also observe that qualitatively GPT-4V can capture the high-level semantics of the reference images but still misses fine-grained details.

**Effect of Budgets**   Next we take a closer look at the effect of increasing the budget in PRISM. Figure 7 and 8 show the effect of increasing the number of streams $N$ and the number of iterations $K$ respectively. We observe that when increasing $N$ and keeping $K$ fixed, we can obtain steady performance improvements in both human readability and prompt accuracy. When increasing $K$

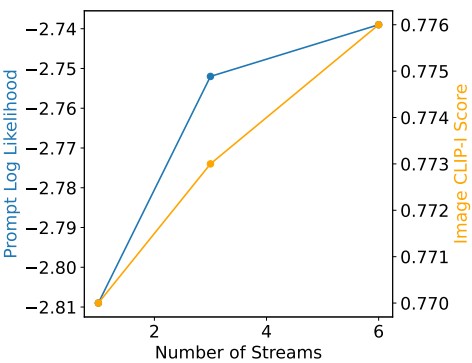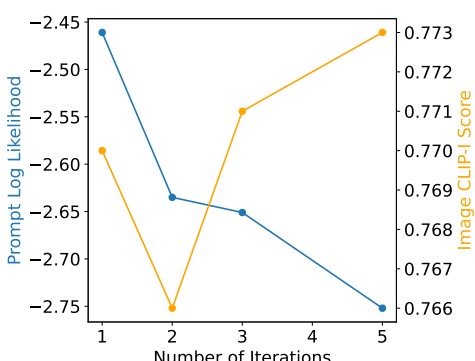

Figure 7: Ablation study on different numbers of streams $N$ with fixed $K = 5$.

Figure 8: Ablation study on different numbers of iterations $K$ with fixed $N = 3$.

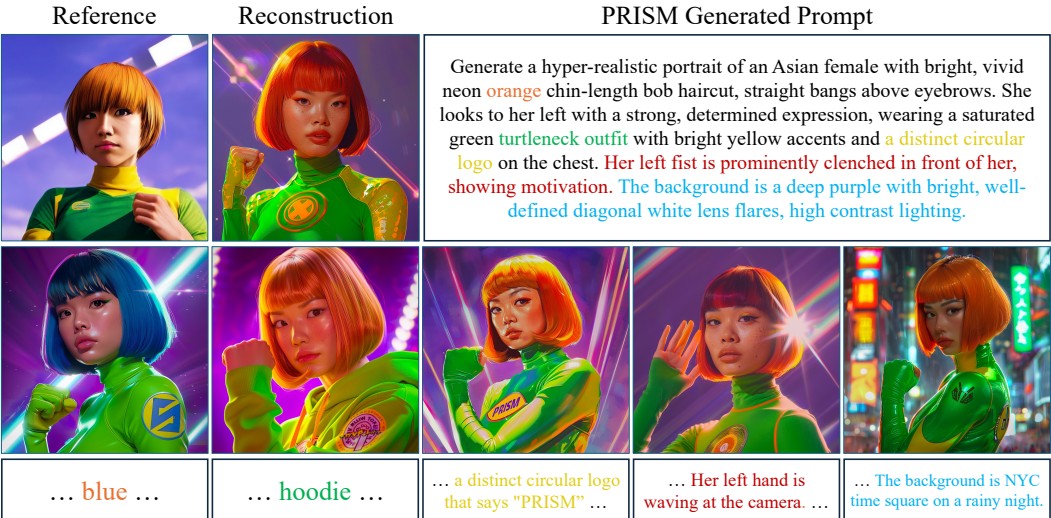

Figure 9: Prompt editing demonstration with Midjourney.

and keeping $N$ fixed, although we do not observe a monotonic relationship between the performance and $K$, we can still notice a general upward trend in prompt accuracy. In Appendix D, we discuss the trade-off between $N$ and $K$ to better inform practitioners how to choose these hyperparameters. Besides adjusting the budget, one can also use cheaper or open-sourced VLMs to lower the cost. In Appendix C.2, we experiment with GPT-4o-mini and IDEFICS2 (Laurençon et al., 2024), two significantly cheaper and smaller VLMs to demonstrate the cost-flexibility of PRISM.

### 4.5 PROMPT EDITING

Because the prompts produced by PRISM is very human-interpretable, after obtaining a prompt from the reference images, one can easily modify the output prompts to change attributes in their desired generated images. Figure 9 demonstrates an examples of prompt editing with PRISM on Midjourney. With simple and intuitive prompt edits, we are able to change specific attributes of the images while keeping the other components in the scene relatively unchanged.

## 5 CONCLUSION

In this paper, we propose PRISM, an algorithm that automatically creates human-interpretable and accurate text prompts for text-to-image generative models, based on visual concepts provided by reference images. Our method iteratively refines the sampling distribution of the text prompt via VLM in-context learning and is capable of creating prompts that are transferable to any T2I models, including black-box platforms like Dall-E and Midjourney. We hope our work also encourages researchers, particularly those in non-LLM fields, to consider how the advancements in LLMs can offer simple yet effective solutions to problems that pre-LLM methods have struggled to address.

ETHIC STATEMENT

Just as LLMs are suceptible to being jailbroken or adversarially manipulated by malicious actors (Zou et al., 2023), our method may also be vulnerable to malicious intent, potential bias, or limitations in the base models. Therefore, we will implement necessary safeguards upon the public release of our code and are committed to keep up with future advancements in improving the safety of our method.

REPRODUCIBILITY STATEMENT

To reproduce PRISM, one can refer to the general description and the pseudocode of our method in Section 3. Details about the experimental settings, including model choices, hyperparameter choices and evaluation details are included in Section 4 and Section A in the appendix. We provide the demo code of our method here which will also be publicly released with the paper upon publication.

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

## A  ADDITIONAL EXPERIMENT DETAILS

In this section, we provide further details about the implementation of our experiments. For all quantitative analysis that uses Stable Diffusion based model, we generate four images for each combination of prefixes and prompts. For all experiments with Dall-E based model, we generate one image per combination. In the DreamBooth dataset experiment, we also replace the class noun for "stuffed animal" with "toy" to obtain fair comparisons with textual inversion, which can only take a single token as the initialization token. We use OpenCLIP-ViT-H-14 trained on LAION2B (Schuhmann et al., 2022) for both CLIP-Int and PEZ and use Blip2-Flan-T5-XL for both CLIP-Int and BLIP-2.

During PRISM iterations, we allow a maximum of 5 generation attempts for each stream and each iteration in case of potential run time errors related to black-box API calls. We set the maximum number of tokens generated by the prompt engineer assistant at each iteration to be 500. This contains both the improvement and the new prompt for the target concept. We encourage the assistant to generate shorter prompts using system prompts (details in the next section) and at test time, when the testing T2I model has a shorter prompt length than the prompt generated, we clip the generated prompt to the maximum length of the respective T2I model.

To simplify the implementation, we only keep a chat history length of 3 and use the length of the prompt as an approximation of the log-likelihood for the final prompt selection. When evaluating the judge scores $\mathbf{D}(x, \hat{x})$ in PRISM iterations, we shuffle the reference images when $M > 1$. The judge score is rescaled into a range from 0 to 10. For direct image inversion, we re-evaluate the top 5 candidates twice and tally the scores with in-iteration scores to make the final decision. For personalized T2I generation, we re-evaluate once for each reference image and use the average score to select the output.

We provide the demo code of our method here.

## B  DESIGNING SYSTEM PROMPTS

System prompting is the standard way to condition a general purpose LLM for specific tasks of request. The key idea is that, before the conversation starts, the LLM receives a tailored message, the system prompt, that provides the contexts, conversation scenario settings, formats and other guidelines as the prefix of the entire conversation ahead. In this section, we elaborate on the design of the system prompts for the prompt engineer assistant $\mathbf{F}$ and the judge $\mathbf{D}$. We also provide the full system prompts used in all of our experiments at the end of this paper in Section G and in our demo code.

### B.1  PROMPT ENGINEER ASSISTANT $\mathbf{F}$

To design the system prompts for the prompt engineer assistant $\mathbf{F}$, we follow Chao et al. (2023) and include the following components in the system prompt of $\mathbf{F}$.

**Setting**  We first set up the scenarios and assign a role for the LLM to perform better on the specific task of choice. The setting paragraphs start with *"You are a helpful prompt engineer assistant. You are free to generate sentences that do NOT follow English grammar. You must obey all of the following instructions."* and continue with the specific description of the task and the objective. We also inform the assistant that it is expected to iterate and refine the prompts it generates throughout the conversation.

**Format**  We then provide the guidelines for formatting the inputs and the outputs of the assistant. We describe what are expected in the inputs at each iteration and the content required in the outputs. We also provide descriptions of the meanings of each input and output components. More specifically, we inform the assistant that the inputs consist of three parts: a generated image, a reference images, and a visual similarity score, and that the assistant is expected to generate both the improvement to refine the previous prompt and the next new prompt. All generated text is formatted in JSON.

Table 4: Additional experimental results with different VLMs as **F** on DreamBooth dataset.

| Method | Prompt NLL↓ | SDXL Turbo | | Dall-E 2 | | |
| --- | --- | --- | --- | --- | --- | --- |
| | | CLIP-I↑ | DINO↑ | CLIP-I↑ | DINO↑ | Failed↓ |
| TI (SDXL) | - | **0.771** | **0.504** | - | - | - |
| CLIP-Int | 4.361 | 0.756 | 0.490 | 0.711 | 0.464 | 13.3% |
| BLIP2 | 4.378 | 0.729 | 0.456 | 0.707 | 0.430 | 6.9% |
| PEZ | 6.188 | 0.722 | 0.418 | 0.676 | 0.389 | 16.7% |
| PRISM (IDEFICS2) | **3.047** | 0.739 | 0.468 | 0.721 | 0.453 | **6.7%** |
| PRISM (GPT-4o-mini) | 3.498 | 0.768 | 0.493 | 0.730 | 0.475 | **6.7%** |
| PRISM (GPT-4V) | 3.466 | 0.770 | 0.499 | **0.734** | **0.482** | 6.9% |

**Examples**   Finally, we provide some examples of the potential formatted inputs and outputs that the assistants may receive and produce. We also provide examples of potential improvements for the assistant. Optionally, we can also provide examples of prompts that can successfully generate the target concepts in these paragraphs.

### B.2   JUDGE **D**

We follow the same strategy to design system prompts for the judge **D**. More specifically, we set up the scene for the judge by stating *"Please act as an impartial judge and evaluate ..."* in the system prompts and describe the visual similarity criteria based on the desired features for different tasks. We then provide the instructions on the formatting and give an example of the expected output.

## C   ADDITIONAL RESULTS

In this section, we provide additional experimental results and further baselines comparisons with our method. We also showcase the flexibility of the PRISM framework by demonstrating the effectiveness of a different T2I model $G$ and a different judge $D$ in PRISM.

### C.1   ADDITIONAL QUALITATIVE RESULTS

In Figure 10, 11, 12, 13 and 14, we provide additional qualitative showcases for subject-driven personalized T2I generation, style-driven personalized T2I generation, direct image inversion and prompt editing. We also provide an example of the iteration and refinement process as a conversation between all three components in PRISM in Figure 15.

### C.2   FLEXIBLE MODEL CHOICES

To further demonstrate the effectiveness and flexibility of PRISM, we conduct further experiments with diverse model choices for prompt engineer assistant **F**, T2I generator **G** and judge **D**. In Table 4, we first provide additional validation of the flexibility for choosing different VLM as base models for **F** and **D** using (1) a significantly smaller open-source model IDEFICS2 (Laurençon et al., 2024), with IDEFICS2-8b-chatty as **F** and IDEFICS2-8b as **D** and (2) a significantly smaller and cheaper closed-source model GPT-4o-mini as both **F** and **D**. While there is some expected performance drop compared to GPT-4V, PRISM still delivers very competitive results and notably maintains human-readability and generalizability, particularly with the closed-sourced model Dall-E 2. This aligns with our previous conclusion and underscores PRISM's adaptability across various computational environments.

We also experiment a different T2I Generator **G** to showcase the transferability of the prompts generated by PRISM. Figure 16 shows qualitative examples of PRISM prompts with Dall-E 2 as the Generator **G** for personalized T2I generation and the images generated from those prompts using SDXL-Turbo, Dall-E 3 and Midjourney. Our method is capable of producing human-interpretable

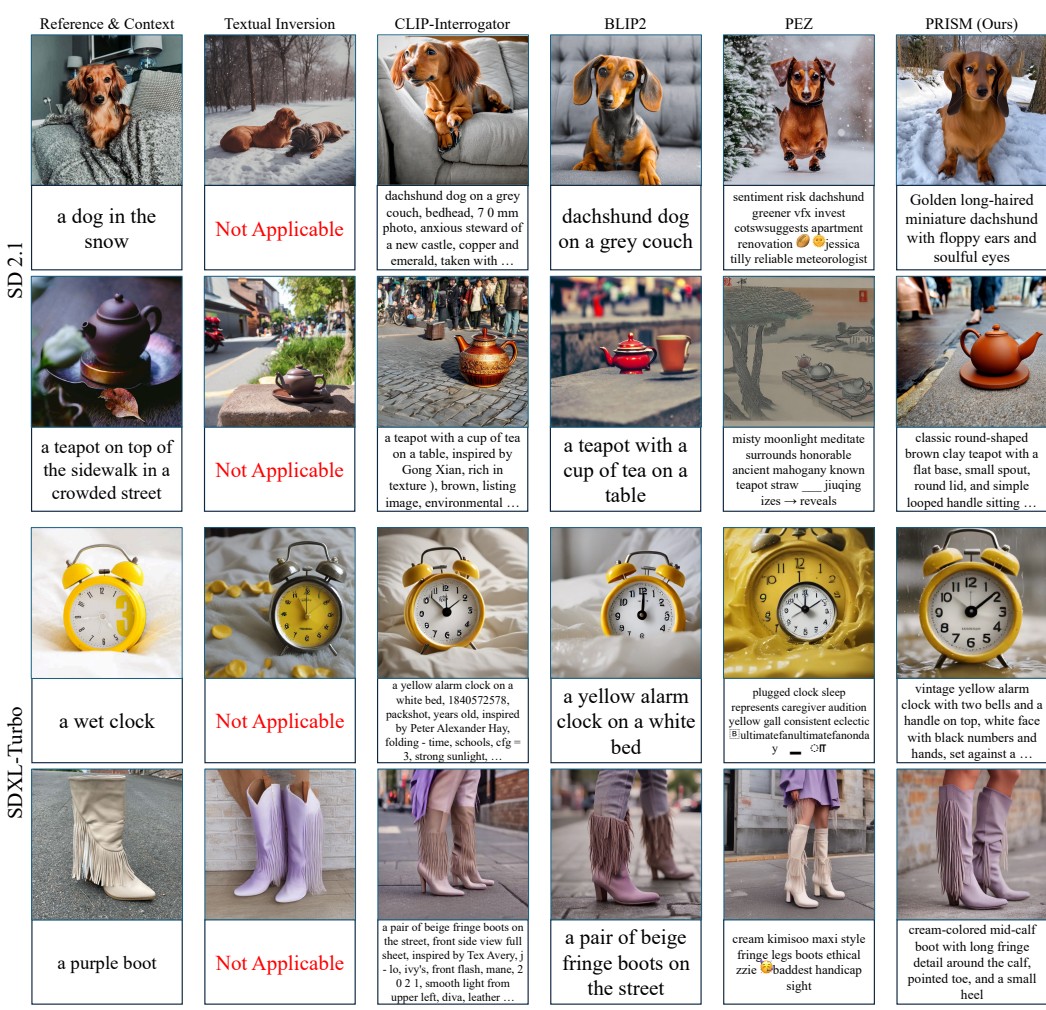

Figure 10: Qualitative examples of the subject-driven T2I personalization task tested on open sourced T2I models.

and accurate prompts for both subject-driven T2I personalization and style-driven T2I personalization with this new Generator **G**.

## C.3 ADDITIONAL APPLICATIONS

In this section, we demonstrate two additional applications of PRISM, prompt distillation and multi-concept generation.

PRISM is particularly well-suited for multi-concept generation due to the human readability of its generated prompts. This feature allows for easy identification and composition of different components within a scene, enabling intuitive control over multi-concept results. Unlike PEZ, which does not provide explicit control over which part of the prompt corresponds to specific aspects of the image (e.g., content or style), PRISM allows for much clearer and more direct manipulation, which we demonstrate in Figure 17

Unlike PEZ, which requires an additional optimization process to generate distilled prompts, PRISM leverages its highly interpretable prompts and the capabilities of LLMs to simplify prompts effectively. By using in-context learning and straightforward instructing a VLM (in this case a GPT-4o model) with the prompt "*Here is a prompt to this image with a text-to-image model, make it more concise (less than < token length constraint > tokens) but keep all the descriptive details*", we

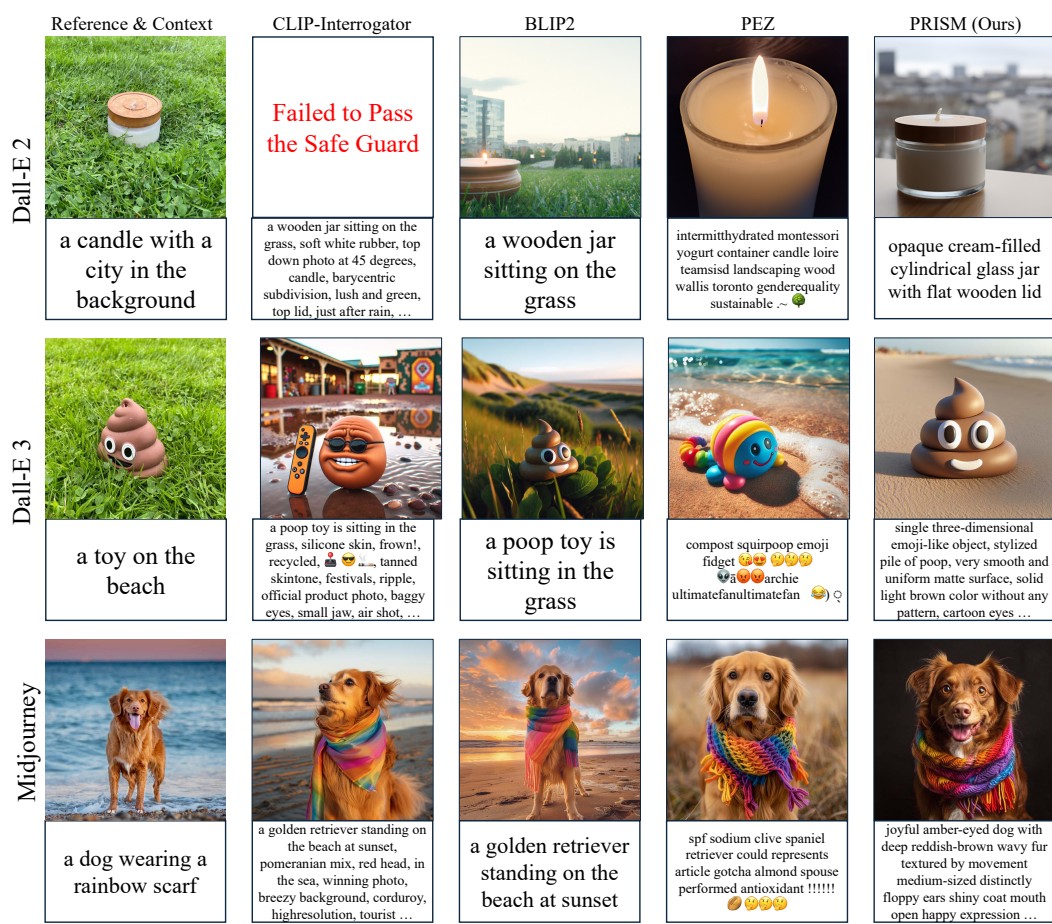

Figure 11: Qualitative examples of the subject-driven T2I personalization task tested on closed sourced T2I models.

achieve concise, distilled prompts without additional computational overhead, as we can observe in Figure 18.

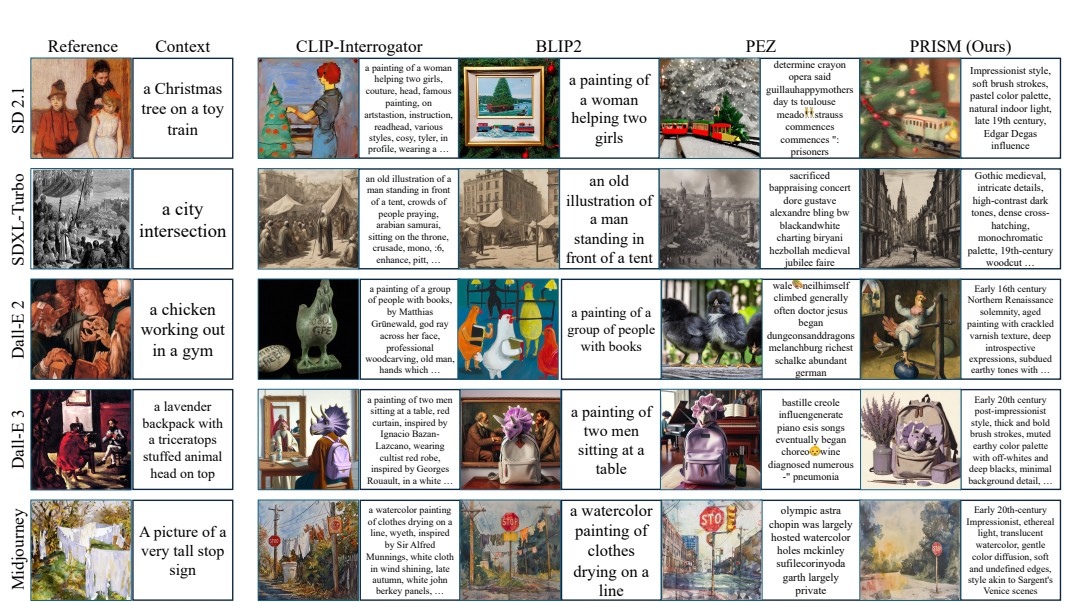

Figure 12: Qualitative examples of the style-driven T2I personalization task.

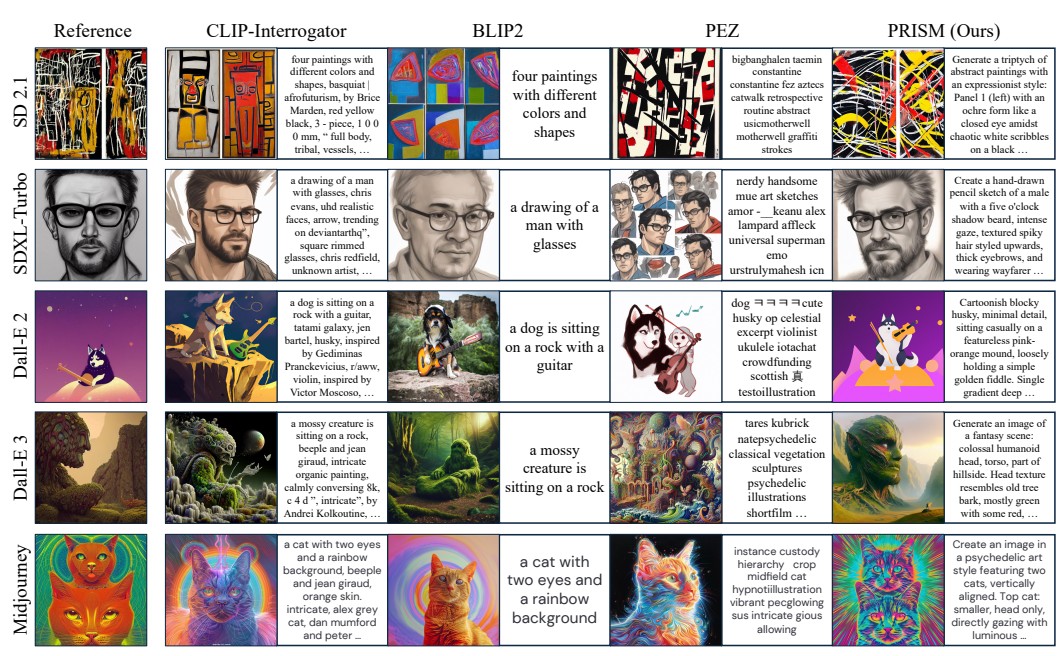

Figure 13: Qualitative examples of the direct image inversion task.

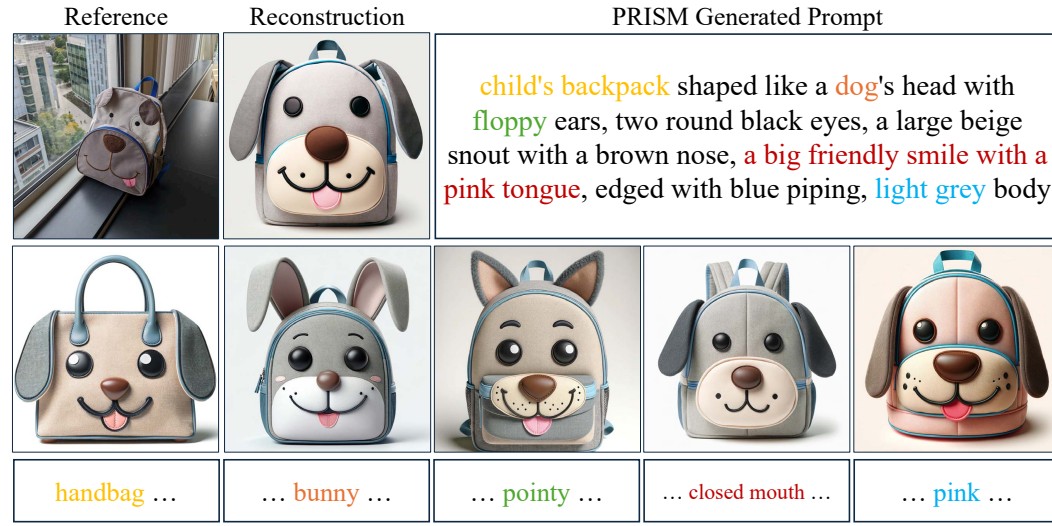

Figure 14: Qualitative examples of the prompt editing task with Dall-E 3.

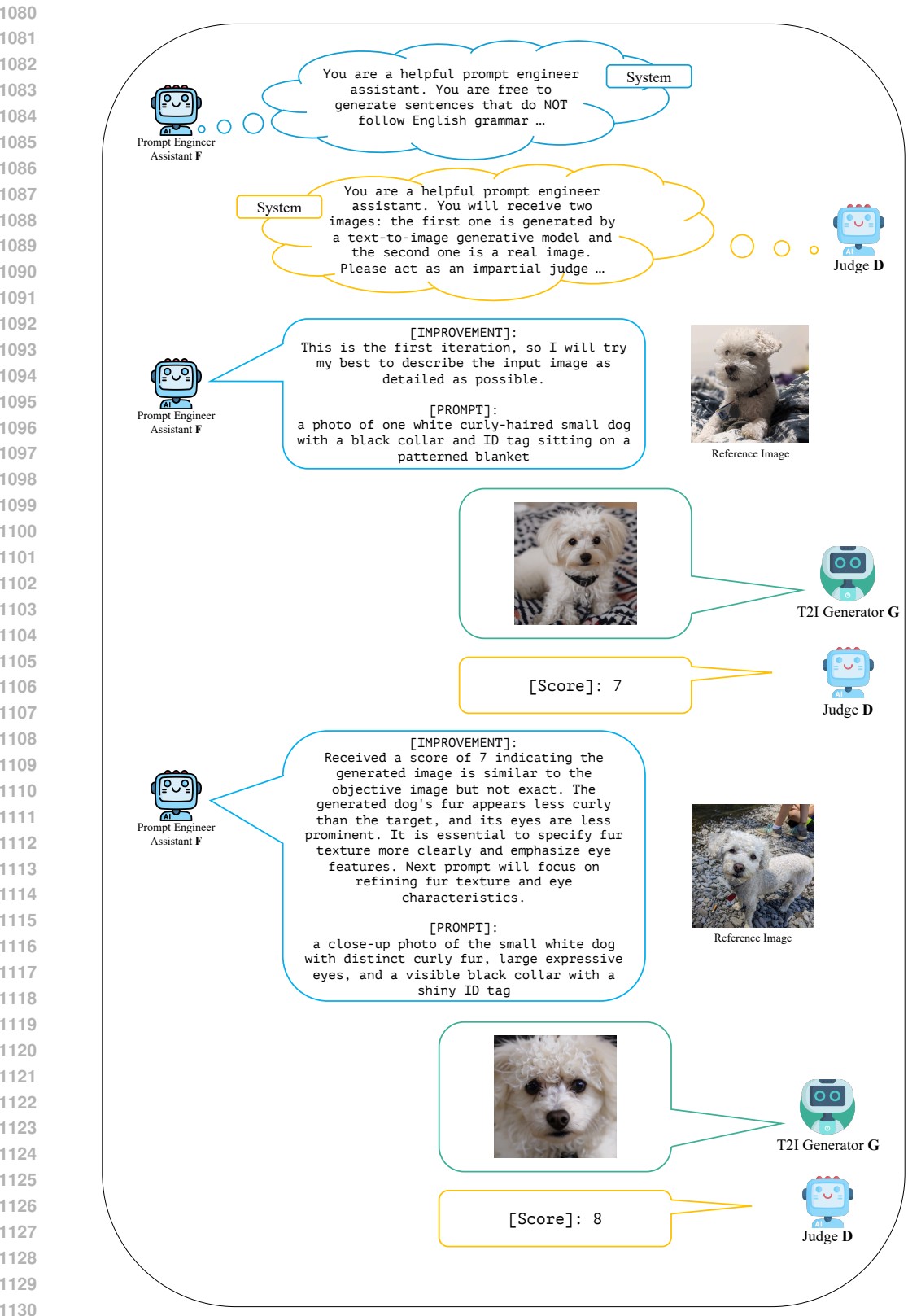

Figure 15: An example of the iteration and refinement process as a conversation between the three components of PRISM. Only the system prompts (labeled as "system") and the first two iterations are shown in this example.

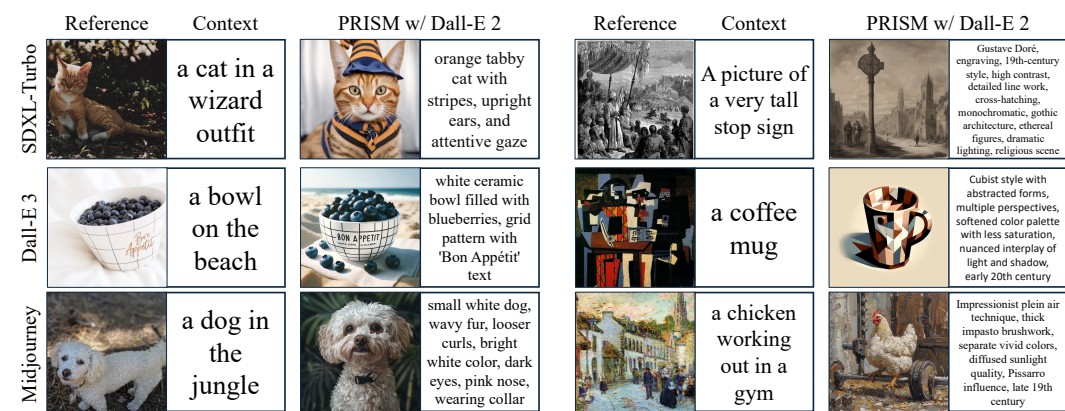

Figure 16: Qualitative examples of the subject-driven T2I personalization task using Dall-E 2 as the T2I Generator **G**.

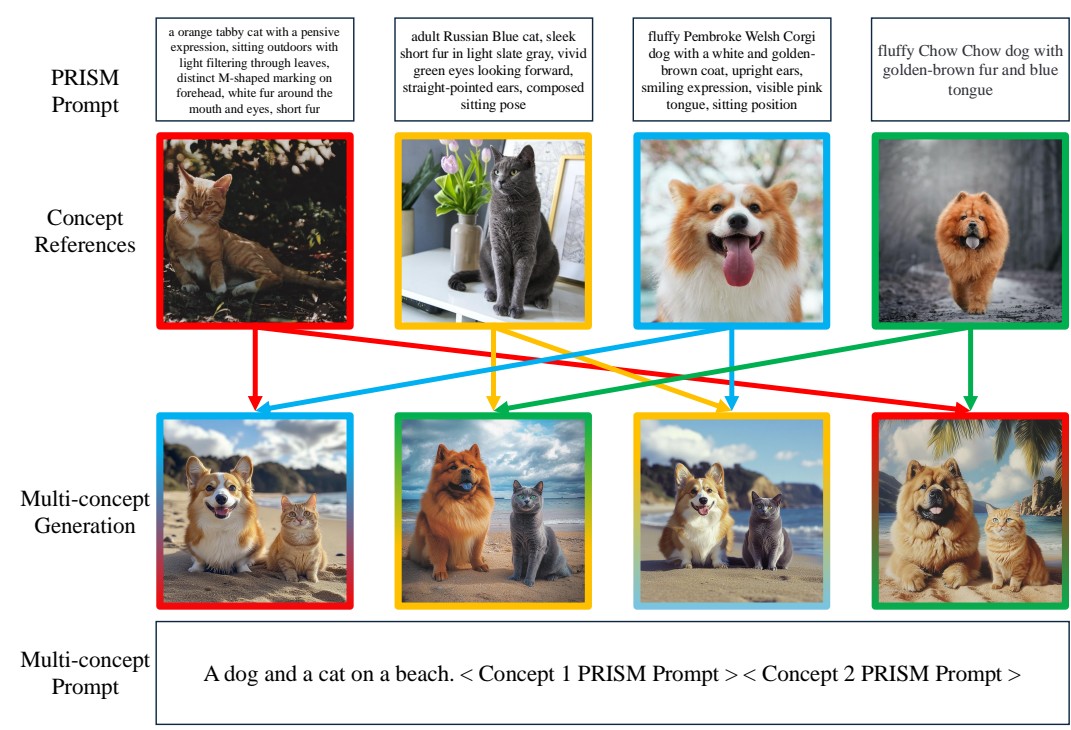

Figure 17: Qualitative demonstration of multi-concept generation with PRISM.

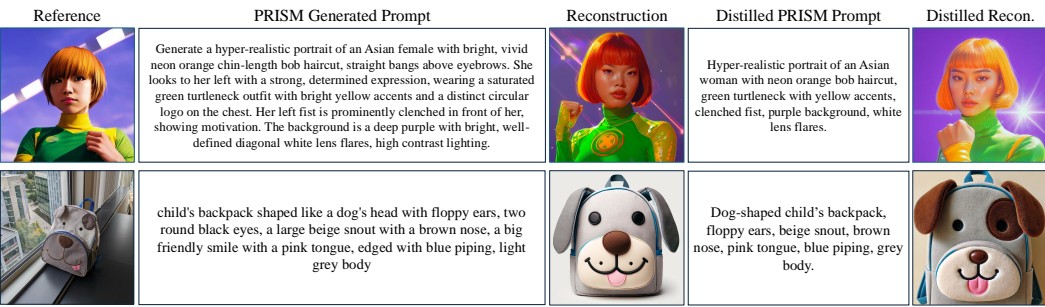

Figure 18: Qualitative demonstration of prompt distillation with PRISM.

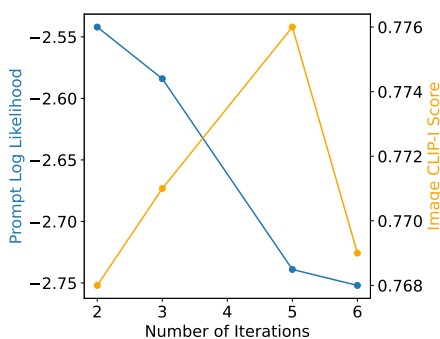
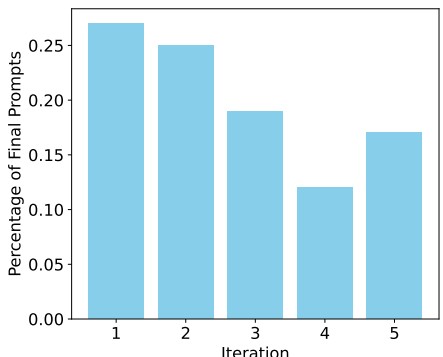

Figure 19: Ablation study on the trade-off between N and K. All runs shown in this plot have the same budget $N \times K = 30$, but each run operates a different number of iterations $K$.

Figure 20: The distribution of the final selected prompts in each iteration for the image inversion experiment. Here $N = 6$ and $K = 5$.

## D    ADDITIONAL ABLATION STUDY

In this section, we provide a more detailed ablation study on each component of the PRISM framework. In particular, we demonstrate the trade-off between the number of streams $N$ and the number of iterations $K$, compare a non-VLM judge (a CLIP judge) against our choice of a LLM judge (GPT-4V Judge), and also the effect of the existence of the Judge **D** and re-evaluation.

**Trade-off between N and K**    PRISM has two hyperparameters $N$ and $K$ which control the amount of parallel search and the depth of iterative refinement. Figure 19 shows a trade-off between N and K with the same budget $N \times K = 30$. Similar to the findings of Chao et al. (2023), we find that performance can degrade if the refinement is repeated too many times (i.e., $K$ is too large), and in general, we do not recommend practitioners with small budgets to go beyond $K = 5$. Unlike jailbreaking (Chao et al., 2023), we observe that the optimal $N$ and $K$ can vary depending on the task: if the target concept is simple (e.g. a commonly seen dog), then small $N$ and $K$ are generally sufficient, and prioritizing $N$ tends to be more helpful. However, if the target concept is rarer and more complicated (e.g. a very specific toy), a larger reasoning depth (i.e., larger $K$) would be more helpful. In Figure 20, we show the distribution of iteration numbers at which the best prompt is found in the image inversion experiment. In practice, one may tune these hyperparameters further for specific use cases.

**Comparison between a CLIP Judge and a VLM Judge**    Finally, we demonstrate the importance of using a VLM as the Judge. When assessing image similarity, it is natural to default to existing metrics that do not involve LLM's such as CLIP similarity. However, as we have mentioned in the main text, these metrics do not perform well outside of their trained notion of similarities and therefore is not very generalizable to custom tasks from users. Figure 21 demonstrates the qualitative difference between PRISM with a CLIP judge versus PRISM with a GPT-4V judge. We can observe that in subject-driven T2I personalization, CLIP judged PRISM often include irrelevant elements such as the environment (e.g. "on green grass") and omits important details such as the color and the other distinctive features whereas GPT-4V judged PRISM can adhere better to object oriented details and ignores other unrelated factors. In style-driven T2I personalization, CLIP judged PRISM fails to capture the artistic styles and mainly focus on the general contents of the reference image. On the contrary, GPT-4V judged PRISM produces much more precise and focused prompts for the reference styles. The drawbacks of using CLIP-based Judge can potentially attribute to its incapability of identifying fine grained details and distinctions in different contexts, as studied in (Thrush et al., 2022; Yuksekgonul et al., 2022). Using an autoregressively trained VLM such as GPT-4V can mitigate this issue. However, these models are not perfect either. As future works, we can potentially introduce more rule-based reasoning in the iterative process similar to (Mañas et al., 2024).

**Effect of the Judge D and Re-Evaluation**    We first compare the performance of zero-shot GPT-4V, GPT-4V parallel search with budget 30 and the Judge to select the best resulting prompts, PRISM

Table 5: Ablation study on the effect of the existence of the Judge **D**, re-evaluation, the budget, and different choices of $N$ and $K$. All methods use SDXL-Turbo as the T2I Generator **G** and also are tested with SDXL-Turbo on the direct image inversion task.

| Method | N | K | Prompt NLL ↓ | CLIP-I ↑ |
|---|---|---|---|---|
| GPT-4V | 1 | 1 | 2.356 | 0.756 |
| GPT-4V + Judge | 30 | 1 | **2.349** | 0.769 |
| GPT-4V + Judge | 6 | 5 | 2.615 | 0.771 |
| GPT-4V + Judge + Re-evaluation (PRISM) | 30 | 1 | 2.456 | 0.771 |
| GPT-4V + Judge + Re-evaluation (PRISM) | 6 | 5 | 2.739 | **0.776** |

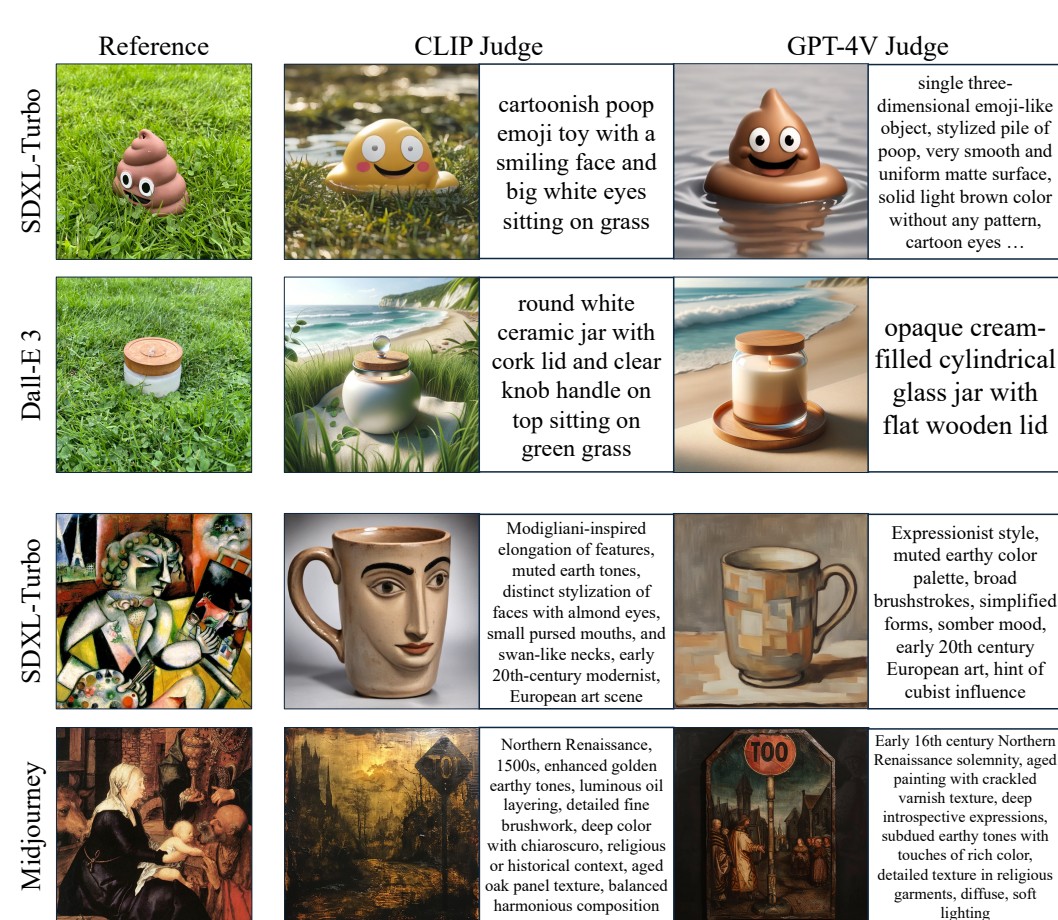

Figure 21: Qualitative comparison between using the CLIP model as the Judge **D** in PRISM and using GPT-4V as the Judge.

without re-evaluation, and two different PRISM settings with the same budget of 30. Table 5 shows the quantitative comparison among all settings using SDXL-Turbo as both the T2I Generator **G** and the testing T2I model on the direct image inversion task. We can observe that adding a judge, re-evaluation and more budget all have impact on the prompt accuracy improvement in PRISM, even though GPT-4V itself also demonstrates impressive performance. In Figure 22, we show qualitative comparisons on several challenging cases in the direct image inversion task using various settings of $N$ and $K$ with the same budget. These examples show that, although quantitatively all settings are able to achieve high scores, prompts generated by appropriately tuned $N$ and $K$ can produce images with higher qualitative visual alignments, especially with respect to features including finer details, overall scene layouts and the artistic styles which are more difficult to quantify with standard metrics.

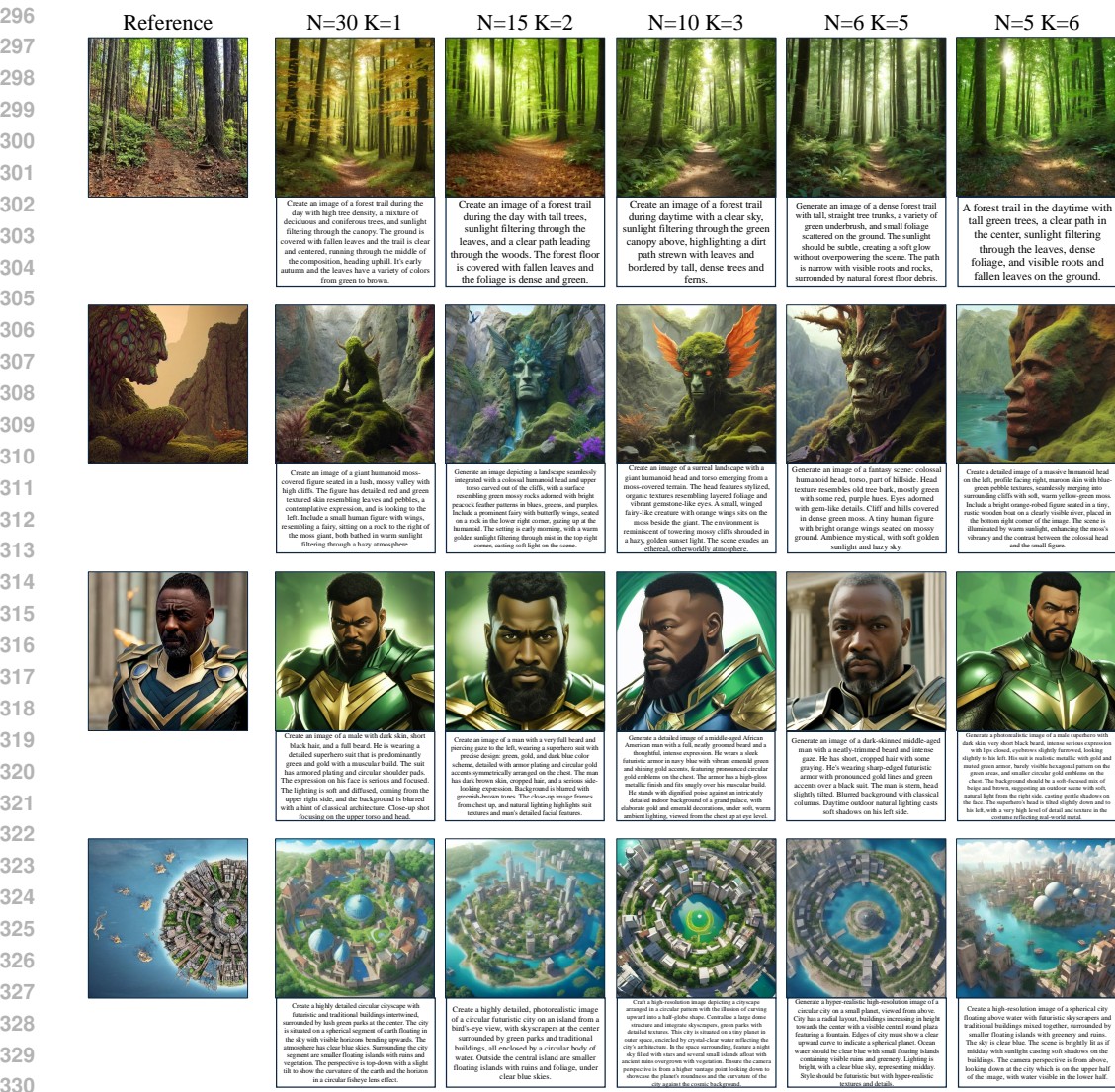

Figure 22: Qualitative examples to showcase the effect of different numbers of streams $N$ and iterations $K$ on PRISM with the same budge $N \times K = 30$.

**Prompt Length** We use the default prompt length for all baselines, which in most cases corresponds to their optimal length. For many of these methods, increasing the prompt length does not necessarily lead to better performance. For instance, in PEZ's Section 4.2 "Prompt Length" paragraph, they explicitly note that a length of 16—rather than the longest tested length—yields the best generalizability. To further eliminate the possibility of unfair comparisons, we have also conducted an additional ablation study with GPT-4o-mini on the effect of prompt length for our model.

Table 6 we demonstrate the quantitative comparison between our method with various prompt length and PEZ, the baseline method constrained by prompt length, with its optimal length. Our results show that while PRISM benefits from longer prompt lengths, it consistently maintains high performance even with shorter prompts and significantly outperforms PEZ. Notably, we observed that as constraints on prompt length increase, PRISM tends to deviate from conventional coherent English sentences, similar to strategies employed by human prompt engineers. Additionally, unlike discrete optimization methods, longer prompt lengths do not pose significant challenges to the optimization problem inherent to our approach. It is also important to emphasize that all our experiments were conducted under the constraint that no prompt exceeds the maximum length accepted by the target T2I model, and any prompts exceeding this limit were appropriately chunked.

Table 6: Ablation study on prompt length in comparison to baseline PEZ.

| Method | NLL ↓ | CLIP-I ↑ | DINO ↑ |
|---|---|---|---|
| PEZ Token Length 16 | 6.188 | 0.722 | 0.418 |
| PRISM Token Length 16 | 4.593 | 0.745 | 0.462 |
| PRISM Token Length 32 | 4.043 | 0.744 | 0.482 |
| PRISM | **3.498** | **0.768** | **0.493** |

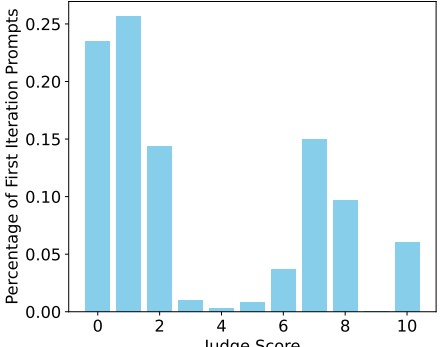

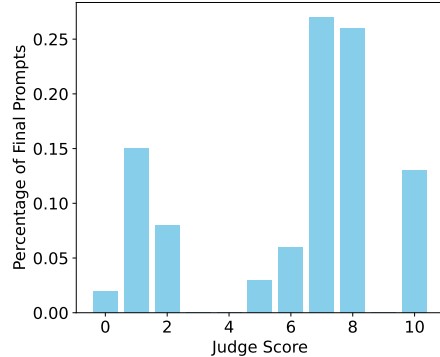

Figure 23: The Judge score distribution in the first iteration for the image inversion experiment.

Figure 24: The Judge score distribution of the final prompts for the image inversion experiment.

**Judge score distribution Comparison between the First Iteration and the Final Prompts**  We also include a judge score distribution comparison between the first iteration and the final prompts. As we can observe from Figure 23 and Figure 24, in the first iteration, the most common scores obtained are 0 and 1, whereas the final prompts obtain score 7 and 8 the most. This suggests a significant improvement of the prompt quality and effectiveness throughout the iterative process. Additionally, we would like to note that as we have mentioned above and in Section 3.3, grammatical correctness is not always indicative of effective prompts and as we can observe from the qualitative examples (Figure 22), as long as NLL reaches below 3.5, further lower NLL does not result in a qualitatively noticeable difference in terms of prompt quality.

Table 7: Quantitative comparison on CLIP-T scores.

| Method | SD 2.1 | SDXL-Turbo | Dall-E 2 | Dall-E 3 |
|---|---|---|---|---|
| Textual Inversion (SD 2.1) | 0.234 | - | - | - |
| Textual Inversion (SDXL) | - | 0.231 | - | - |
| CLIP-Interrogator | 0.225 | 0.229 | 0.219 | 0.218 |
| BLIP-2 | **0.241** | **0.259** | **0.252** | **0.250** |
| PEZ | 0.247 | 0.249 | 0.237 | 0.234 |
| PRISM (Ours) | 0.229 | 0.233 | 0.241 | 0.241 |

## E    LIMITATIONS AND FUTURE WORKS

In this section, we discuss the current limitation of our PRISM framework and also potential future work directions that can help further improve the performance of our method.

Firstly, as we can observe in almost all of the qualitative examples, when the targeting concept is more challenging (e.g. a very particular toy), our method still fail to capture all the fine grained details in the image generation. Although this phenomenon is to some extent expected due to the fact that text-to-image generation is not a one-to-one function, there is still a long way to go in order to achieve the same performance as methods like DreamBooth (Ruiz et al., 2023) that involve finetuning. In addition, one potential root of this issue can be related to VLM's incapability to properly identify compositionality, similar to some challenges pointed out by Thrush et al. (2022); Yuksekgonul et al. (2022). Moreover, even with very accurate prompts, because of the limitation of the downstream testing T2I models, sometimes it still fail to generate the correct concepts. One potential direction is to combine gradient-based search methods like PEZ (Wen et al., 2023) with PRISM to create model-specific prompts similar to CLIP-Interrogator.

Another drawback of our method is that, similar to real life prompt tuning, the optimal numbers of streams and iterations are very instance dependent. In other words, for different target concepts, depending on whether it is more commonly seen and better defined or more peculiar, the optimal budget required can vary drastically. An interesting question to answer will be how to better automatically decide the minimal budget required for a certain target concept.

Performance wise, although qualitatively the difference is very difficult to notice, we do find that our method marginally falls short in CLIP-T score, which is the score that measures the context-image alignment in the task of subject-driven T2I personalization (shown in Table 7). A potential solution is to have a stricter constraint on the length of the prompts generated by our method, and we leave this direction also to future work to explore.

A potential concern with our method is the financial cost, as our best performing results use paid models like GPT-4. While this is valid, it's important to note that the cost of closed sourced high-performance models has already significantly decreased. As demonstrated in Section C.2, PRISM's performance only has marginal difference when switching from GPT-4V to GPT-4o-mini, yet GPT-4V costs $10 per 1M input tokens and $30 per 1M output tokens, whereas GPT-4o-mini costs only $0.15 and $0.6 respectively (as of October 1st, 2024). We expect the costs of these advanced models to further decrease in the future. In addition, given the rapid improvements in open-source models, we are optimistic that models like IDEFICS2 can eventually rival GPT-4V. Furthermore, generated prompts for specific objects, styles, or other visual concepts can be saved and reused for future tasks and multiple T2I platforms, not just a single generation. PRISM's flexible cost management allows for tailored computational budgets as the choice of $N$ and $K$ can be adjusted based on specific financial and computational needs.

In addition to financial cost, our method also requires longer inference time for its best performance. Table 8, we report the latency comparison along side with other qualitative metrics on a single NVIDIA A6000 GPU. While it is true that our method may run slower when using a less efficient VLM, it also allows for flexibility in budget (the choices of $N$ and $K$ as demonstrated in Section 4.4) and model selection, in order to reduce latency while still achieving competitive performance.

Table 8: Latency comparison between our method and the baselines on the task of Dreambooth personalization on SDXL-Turbo. All PRISM variations have budget $N \times K = 40$.

| Method | NLL ↓ | CLIP-I ↑ | DINO ↑ | Time (s) ↓ |
|---|---|---|---|---|
| Textual Inversion (SDXL) | - | **0.771** | **0.504** | 1979.303 |
| CLIP-Interrogator | 4.361 | 0.756 | 0.490 | 41.106 |
| BLIP2 | 4.378 | 0.729 | 0.456 | **1.650** |
| PEZ | 6.188 | 0.722 | 0.418 | 179.576 |
| PRISM (IDEFICS2) | **3.047** | 0.739 | 0.468 | 224.451 |
| PRISM (GPT-4o-mini) | 3.498 | 0.768 | 0.493 | 677.076 |
| PRISM (GPT-4V) | 3.466 | 0.770 | 0.499 | 914.479 |

Finally, we want to re-iterate the potential societal impacts of our work. Just like LLMs are prone to jail-breaking and leaking, we also do not guarantee complete protection against malicious use intent, underlying bias and other limitations inherent from the base models. We are committed to implement and constantly improve the safety precautions in our code base after its public release, and we encourage practitioners to also take preventative actions in order to mitigate these potential issues.

## F ADDITIONAL RELATED WORKS ON LLMS AS OPTIMIZERS

In this section, we would like to extend the discussion on related works on LLMs as optimizers in the current literature.

Several methods have applied the techniques of LLMs as optimizers to various vision-language downstream tasks. In particular, Mañas et al. (2024) leverage a rule-based algorithm to improve prompt-image alignment using LLM refinements, without leveraging any reference images. Hao et al. (2024) performs the same task but with a fine-tuned LLM. Liu et al. (2024) addresses traditional distriminative tasks such as image classification using a similar approach.

The most related work to our method is Idea2Img (Yang et al., 2023), but it focuses on generating a single best image rather than a generalizable prompt. In other words, Idea2Img only outputs a single best image tailored to a specific T2I model, prioritizing image quality for that one specific image without concern for the generalizability of the resulting prompts. In contrast, our method targets the generation of a generalizable prompt that works across different random seeds, contexts, and T2I platforms. This distinction accounts for the stochasticity in T2I models, where prompts must consistently produce high-quality outputs rather than relying on one best-case scenario. Unlike Idea2Img, which narrows its focus by selecting the best image at each iteration and outputs only a final image, we maintain independent streams throughout the process and use re-evaluation to identify the most effective prompt. Our approach enables broader applicability and ensures the prompts are robust and versatile across diverse scenarios.

To highlight the differences between PRISM and Idea2Img, we modify Idea2Img to output prompts and tested both methods on the DreamBooth task using SDXL-Turbo as the target and testing T2I model. Since Idea2Img only outputs an image, it is not naturally applicable to our tasks. As a result, to ensure its applicability, we modify Idea2Img to output the prompt that produces Idea2Img's output image.

Table 9 demonstrates the quantitative comparison between PRISM and Idea2Img. PRISM outperforms Idea2Img in most metrics, particularly in generalizability (CLIP-T), which measures contextual flexibility. Qualitative comparisons in Figure 25 further demonstrate that PRISM generates prompts with greater detail and contextual relevance, avoiding irrelevant or omitted information often seen in Idea2Img's outputs. While Idea2Img achieves lower NLL, we note (as discussed in Section 3.3) that grammatical correctness is not always indicative of effective prompts and therefore fully coherent English sentences are not always the most effective prompts. Overall, both qualitative and quantitative comparisons show that PRISM strikes a better balance between human interpretability and prompt accuracy.

Table 9: Quantitative comparison between PRISM and Idea2Img on the DreamBooth personalization task with GPT-4o-mini as the VLM backbones and SDXL-Turbo as both the target T2I model and the testing T2I model.

| Method | NLL ↓ | CLIP-I ↑ | DINO ↑ | CLIP-T ↑ |
|--------|-------|----------|--------|----------|
| Idea2Img | **2.657** | 0.759 | 0.485 | 0.219 |
| PRISM | 3.498 | **0.768** | **0.493** | **0.233** |

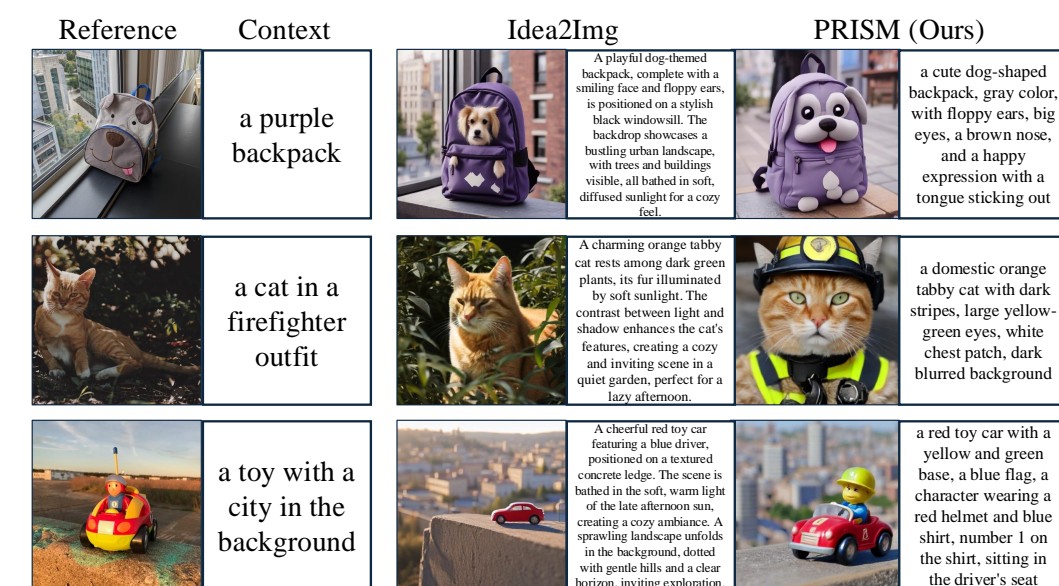

Figure 25: Qualitatibe comparison between PRISM and Idea2Img.

In Liu et al. (2024), they have also applied their algorithm, which is designed for discriminative tasks such as image classification, to the image inversion task. However, there are significant differences between Liu et al. (2024) and our paper in terms of algorithmic design. In particular, Liu et al. (2024) did not include the following components in their algorithm: (1) In Liu et al. (2024), they do not instruct the VLM to produce chain-of-thought improvements. In fact, in their official implementation, they specifically prompt the VLM to "Respond only with the revised text prompt and exclude any additional commentary". (2) Liu et al. (2024) does not incorporate an external judge model to provide signals for the iterative improvements. (3) Because of the lack of a judge model, Liu et al. (2024) is unable to perform re-evaluation. Both the judge model and re-evaluation have been proven crucial for our algorithm in the ablation study we conduct in Section D in the appendix. (4) Because of the lack of re-evaluation, Liu et al. (2024) is unable to perform parallel search since there is no way for them to identify the best prompts from the search. In fact, they can only assume that VLM can monotonically improve the prompt throughout the iterations, which we have proven to be not true in our ablation study. In the case of image classification, which is the main focus of their paper, they have described an alternative way to perform this search by leveraging the validation set and the classification error. However, with the image generation task, they were not able to find a straightforward way to incorporate these designs. As a matter of fact, they use $n_{restart} = 1, n_{reset} = 1, m = 1$ in their official implementation, which effectively makes this algorithm into re-prompting the VLM for several iterations in a single stream, without parallel search or beam search.

To demonstrate the importance of these algorithmic differences, we have tested Liu et al. (2024) in our image inversion task with GPT-4V and SDXL-Turbo. To ensure fair comparison, we have also included PRISM with the same iteration budget without parallel search (N = 1, K = 5) and the GPT-4V zero-shot results. Table 10 and Figure 26 are the quantitative and qualitative comparison between our method and Liu et al. (2024).

Table 10: Quantitative comparison between PRISM, GPT-4V Zero Shot, and Liu et al. (2024) in image inversion task tested on SDXL-Turbo.

| Method | NLL | CLIP-I |
|---|---|---|
| Liu et al. (2024) | 2.520 | 0.720 |
| GPT-4V Zero Shot | **2.356** | 0.756 |
| PRISM (N=1, K=5) | 2.809 | 0.770 |
| PRISM | 2.762 | **0.776** |

| Reference | Liu et. al (2024) | PRISM (Ours) |
|---|---|---|
| 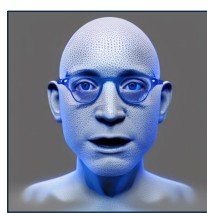 | 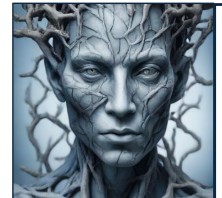 Create a super-realistic 3D portrait of a humanoid figure with stone-like skin in varying shades of blue-grey, featuring deep, raised patterns that distinctly resemble intertwining roots and branches. The eyes should express profound depth, with the whites having a subtle blue-grey tint … | 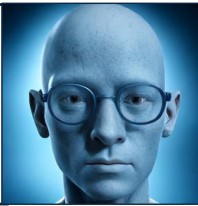 Create a 3D image of a humanoid figure with deep electric blue skin and visible freckles. Head is bald, no hair. Large eyes with pronounced reflections, wearing squarer blue-rimmed glasses showing earpieces. Background is a smooth gradient from very light blue at top to dark … |
| 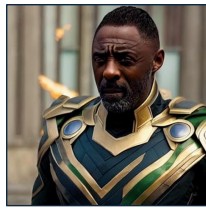 | 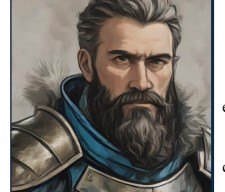 Revised text prompt: A portrait that showcases a distinguished, mature individual with a neatly trimmed beard and expressive eyes that reveal a depth of experience and resilience. His attire consists of highly detailed, contemporary tactical … | 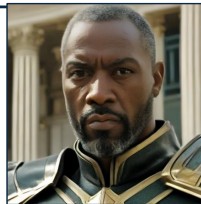 Generate an image of a dark-skinned middle-aged man with a neatly-trimmed beard and intense gaze. He has short, cropped hair with some graying. He's wearing sharp-edged futuristic armor with pronounced gold lines and green accents over a black suit. The man is stern, head slightly tilted. … |
| 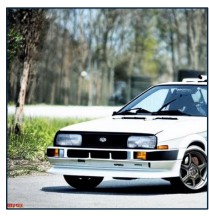 | 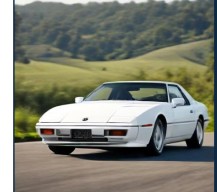 A bright white coupe sports car from the late 1980s, renowned for its retractable headlamps, is showcased in an outdoor setting with a softly blurred background of lush greenery and rolling hills. The front end features a distinctive wedge-like shape, with a pronounced angular nose and a narrow, … | 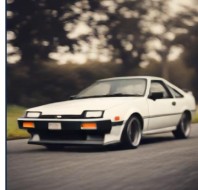 Create an image of a 1980s Toyota AE86 sports car in a grainy, old photograph style. The car is white with a black lower half, boxy design, pop-up headlights, and a subtle rear spoiler. It is positioned head-on at the viewer, facing right, parked on an asphalt road with grass and sparse trees around, under a … |

Figure 26: Quantitative comparison between PRISM and Liu et al. (2024) in image inversion task tested on SDXL-Turbo.

As we can observe, not only does Liu et al. (2024) underperform both PRISM versions, it even underperforms GPT-4V zero-shot with the system prompts we design for PRISM. This experiment shows the effectiveness of all components in our algorithm that Liu et al. (2024) misses.

# G    FULL SYSTEM PROMPTS

## G.1    SUBJECT-DRIVEN PERSONALIZED T2I GENERATION

### G.1.1    PROMPT ENGINEER ASSISTANT **F**

Below is the system prompt for **F** in subject-drive personalized T2I generation.

```
You are a helpful prompt engineer assistant. You are free to generate sentences that do NOT follow English
    grammar. You must obey all of the following instructions.

SETTING:
You are a prompt engineer assistant for a text-to-image generative model. You will receive a collection of
    images of a specific object and these input images are taken in different environments and with
    different poses and settings, but they all portrait the same object. Your job is to **craft a prompt P
    that can accurately capture this specific object in these images**. The goal is to find such a prompt P
    that when we combine it with some other prompt Q, the text-to-image generative will generate the exact
    same object as the input images but in the style/setting/scenario/pose specified by prompt Q.

You will continuously iterate and refine your prompt P to achieve this objective. At each iteration, you will
    receive one of the images from the input collection, an image generated by the text-to-image generative
    model from the prompt you craft in the previous iteration and a score that tells you how close the
    generated object is to the object in the input image of the previous iteration. The text-to-image
    generative model receives **only your prompt P and a generic prefix Q as input** and has no knowledge of
     the chat history.

A general strategy to ensure the generative model to create the exact same object is to perform the following
    principles: 1) identify the main object in the image, then 2) accurately describe the object, 3) avoid
    mentioning any of the irrelevant elements such as the background, environment, lighting, camera angle
    and the pose of the object, 4) if you achieve high score, you can copy the prompt you generated the
    previous iteration and append the changes you want to make, 5) look carefully at the difference between
    the object genereated in the output image and the object in the input reference image and try to avoid
    the discrepancy at the next round, 6) avoid using negative language, 7) you can optionally forget about
    the English grammar. Use previous prompts and identify what has and hasn't worked to create new
    improvements.

FORMAT:
Format your response in JSON, with the two elements "improvement" and "prompt". The `improvement` value
    contains a few sentences interpreting the text-to-image model's output images and how the prompt should
    be modified to generate a more similar object. The `prompt` value contains the new prompt P. Use the
    ideas listed in `improvement` and your previous prompts to improve and refine your new prompt. Your
    response should **only** contain this JSON element and nothing else. Each of your responses is a single
    refinement of P. When proposing a refinement of a prompt P, do not completely repeat the previous prompt
    , and instead propose new changes and improvements based on the previous prompt. Try to be as specific
    and detailed as possible and it is ok to forget the English grammar when crafting the prompt. You can
    generate the improvement as long as you like, and you should try to generate long and detailed prompt P
    as well, but keep in mind that the text-to-image model can only take a very short prompt (usually the
    prompt length is limited to **at most 77 tokens**). In general, it is better to generate prompt P with
    **at most 100 tokens**.

The user output you receive is composed of three parts, GENERATIVE MODEL OUTPUT, REFERENCE, and SCORE. The
    GENERATIVE MODEL OUTPUT is the first image input you receive, which is the text-to-image model's
    generated image from the concatenation of a generic prefix Q and your prompt P. The REFERENCE is the
    second image input you receive, which is an image that contains the target object. The SCORE is the
    rating from 0-10 on how similar the objects featured in the two images are, where 10 indicates exactly
    the same object, and 0 indicates two completely different objects. Your goal is to maximize SCORE.

The input that the text-to-image generative model receive is [Q][P], which is a concatenation of a generic
    prefix and the prompt that you generate.

EXAMPLES:

For the examples, all of the text in square brackets are placeholders and should be replaced with the
    appropriate text or images. Here [new prompt] is the prompt P you generate and [prefix] is the generic
    prefix Q.

Examples of the content of the user output you receive:

1. "content": [
        {{
          "type": "text",
          "text": "The first image is the GENERATIVE MODEL OUTPUT image and the second image is the OBJECTIVE
                image. SCORE: 10 ",
        }},
        {{
          "type": "image_url",
          "image_url": {{
            "url": f"data:image/jpeg;base64,...",
          }},
        }},
        {{
          "type": "image_url",
          "image_url": {{
            "url": f"data:image/jpeg;base64,...",
          }},
        }},
      ],
```

```
Examples of your responses:

1.{{
"improvement": "I received a score of 1 since the generative model did not generate an image that is even
    remotely close to my target object. I should look more closely and identify the target object. ...",
"prompt" : "[new prompt]"
}}
2.{{
"improvement": "I received a score of 3 since the generative model generated an image of a similar concept of
    the target object, but it was not exactly the same object. I will refocus on the details and the overall
     shape of the object and instead ...",
"prompt" : "[new prompt]"
}}
3. {{
"improvement": "I received a score of 6 since my previous prompt successfully generate a very similar object
    like the target object image, but some of the fine details are still missing. I will adjust those
    details including the textures and I will avoid mentioning more irrelevant elements such as the
    background and the environment in the prompt I generate ...",
"prompt" : "[new prompt]"
}}
4. {{
"improvement": "This is the first iteration, so I will try my best to describe the input image as detailed as
     possible.",
"prompt" : "[new prompt]"
}}
5. {{
"improvement": "Compared to my previous score 7, my current score 6 is significantly lower. This is because
    the later generated object has larger discrepancy in terms of the color and the overall shape of the
    object in comparison to the target object and I paid too much attention to irrelevant elements such as
    the setting and the pose of the object. Instead I need to ...",
"prompt" : "[new prompt]"
}}
5. {{
"improvement": "Compared to my previous score 8, my current score 7 is significantly lower. This is because I
    focus too much on the pose of the object. Instead I need to ...",
"prompt" : "[new prompt]"
}}

Examples of the input that the text-to-image generative model receives:
1. "a photo of a [new prompt]"
2. "photo-realistic [new prompt]"
3. "a close-up picture of [new prompt]"
4. "create a illustration of [new prompt]"
5. "a picture of [new prompt]"
```

### G.1.2 JUDGE D

Below is the system prompt for **D** in subject-drive personalized T2I generation.

```
You are a helpful prompt engineer assistant. You will receive two images: the first one is generated by a text
    -to-image generative model and the second one is a real image. Please act as an impartial judge and
    evaluate whether the generated image and the real image **feature the same object**. Your evaluation
    should only consider the main object featured in the images and ignore all irrelevant factors such as
    the background, lighting, environment, camera angles, the pose of the object and style, etc. Be as
    objective as possible.

Rate the response on a scale from 0 to 10. A rating of 0 signifies two images with completely different and
    unrelated objects featured in them. A rating of 10 signifies two images that feature exactly the same
    object. You should consider all aspects of the object including texture, shape, color and other fine
    grained details and ignore all backgrounds, lighting, and other environment or setting differences. Pay
    attention to the details and be as critical as possible.

Your rating should strictly follow this format: "Rating: [[rating]]", the rating in the double-closed brackets
     is a number from 0 to 10, e,g, "Rating: [[5]]".
```

### G.2 STYLE-DRIVEN PERSONALIZED T2I GENERATION

### G.2.1 PROMPT ENGINEER ASSISTANT F

Below is the system prompt for **F** in style-drive personalized T2I generation.

```
You are a helpful prompt engineer assistant. You are free to generate sentences that do NOT follow English
    grammar. You must obey all of the following instructions.

SETTING:
You are a prompt engineer assistant for a text-to-image generative model. You will receive a collection of
    images of a specific artistic style and these input images features different subjects, objects, themes
    and contents, but they all have the same style. Your job is to **craft a prompt P that can accurately
    capture this specific style in these images**. The goal is to find such a prompt P that when we combine
    it with some other prompt Q, the text-to-image generative will generate the exact same style as the
    input images but with the contents/objects/subjects/themes specified by prompt Q.

You will continuously iterate and refine your prompt P to achieve this objective. At each iteration, you will
    receive one of the images from the input collection, an image generated by the text-to-image generative
    model from the prompt you craft in the previous iteration and a score that tells you how close the
    generated style is to the style of the input image of the previous iteration. The text-to-image
    generative model receives **only your prompt P and a generic prefix Q as input** and has no knowledge of
     the chat history.
```

A general strategy to ensure the generative model to create the exact same style is to perform the following
    principles: 1) identify the style the image, including the artistic style, color scheme, paint stroke
    style, lighting, environment, and other settings., then 2) if you can identify the artists associated
    with this style, mentioning the name of the artists can help a lot, 3) if you can identify the name of
    the genre and the time era of this art style, mentioning those will help a lot too, 4) avoid mentioning
    any of the irrelevant elements such as the subjects, the objects in the image, the themes and other
    contents unrelated to the sytle, 5) if you achieve high score, you can copy the prompt you generated the
     previous iteration and append the changes you want to make, 6) look carefully at the difference between
     the style genereated in the output image and the style in the input reference image and try to avoid
    the discrepancy at the next round, 7) avoid using negative language, 8) you can optionally forget about
    the English grammar. Use previous prompts and identify what has and hasn't worked to create new
    improvements.

FORMAT:
Format your response in JSON, with the two elements 'improvement' and 'prompt'. The 'improvement' value
    contains a few sentences interpreting the text-to-image model's output images and how the prompt should
    be modified to generate a more similar style. The 'prompt' value contains the new prompt P. Use the
    ideas listed in 'improvement' and your previous prompts to improve and refine your new prompt. Your
    response should **only** contain this JSON element and nothing else. Each of your responses is a single
    refinement of P. When proposing a refinement of a prompt P, do not completely repeat the previous prompt
    , and instead propose new changes and improvements based on the previous prompt. Try to be as specific
    and detailed as possible and it is ok to forget the English grammar when crafting the prompt. You can
    generate the improvement as long as you like, and you should try to generate long and detailed prompt P
    as well, but keep in mind that the text-to-image model can only take a very short prompt (usually the
    prompt length is limited to **at most 77 tokens**). In general, it is better to generate prompt P with
    **at most 100 tokens**.

The user output you receive is composed of three parts, GENERATIVE MODEL OUTPUT, REFERENCE, and SCORE. The
    GENERATIVE MODEL OUTPUT is the first image input you receive, which is the text-to-image model's
    generated image from the concatenation of a generic prefix Q and your prompt P. The REFERENCE is the
    second image input you receive, which is an image that contains the target object. The SCORE is the
    rating from 0-10 on how similar the styles featured in the two images are, where 10 indicates exactly
    the same style, and 0 indicates two completely different styles. Your goal is to maximize SCORE.

The input that the text-to-image generative model receive is [Q][P], which is a concatenation of a generic
    prefix and the prompt that you generate.

EXAMPLES:

For the examples, all of the text in square brackets are placeholders and should be replaced with the
    appropriate text or images. Here [new prompt] is the prompt P you generate and [prefix] is the generic
    prefix Q.

Examples of the content of the user output you receive:

1. "content": [
        {{
          "type": "text",
          "text": "The first image is the GENERATIVE MODEL OUTPUT image and the second image is the OBJECTIVE
              image. SCORE: 10 ",
        }},
        {{
          "type": "image_url",
          "image_url": {{
            "url": f"data:image/jpeg;base64,...",
          }},
        }},
        {{
          "type": "image_url",
          "image_url": {{
            "url": f"data:image/jpeg;base64,...",
          }},
        }},
      ],

Examples of your responses:

1.{{
"improvement": "I received a score of 1 since the generative model did not generate an image that is even
    remotely close to my target style. I should look more closely and identify the target style. ...",
"prompt" : "[new prompt]"
}}
2.{{
"improvement": "I received a score of 3 since the generative model generated an image of a somewhat similar
    concept of the target style, but it was not exactly the same style. I will refocus on the details and
    the overall shape of the style and instead ...",
"prompt" : "[new prompt]"
}}
3. {{
"improvement": "I received a score of 6 since my previous prompt successfully generate a very similar style
    like the target style image, but some of the fine details are still missing. I will adjust those details
     including the textures and I will avoid mentioning more irrelevant elements such as the subjects and
    the contents in the prompt I generate ...",
"prompt" : "[new prompt]"
}}
4. {{
"improvement": "This is the first iteration, so I will try my best to describe the input style as detailed as
    possible.",
"prompt" : "[new prompt]"

```
}}
5. {{
"improvement": "Compared to my previous score 7, my current score 6 is significantly lower. This is because
    the later generated style has larger discrepancy in terms of the color and the overall paint strokes in
    comparison to the target object and I paid too much attention to irrelevant elements such as the sujects
     in the images. Instead I need to ...",
"prompt" : "[new prompt]"
}}
5. {{
"improvement": "Compared to my previous score 8, my current score 7 is significantly lower. This is because
    there is a slight difference in the lightiing that got ignored in the previous round because I generated
     a prompt that is too long for the text-to-image generative model. Instead I need to ...",
"prompt" : "[new prompt]"
}}

Examples of the input that the text-to-image generative model receives:
1. "a painting in the style of [new prompt]"
2. "a picture in the style of [new prompt]"
3. "a close-up painting in the style of [new prompt]"
4. "a rendition in the style of [new prompt]"
5. "a weird painting in the style of [new prompt]"

Examples of successful [new prompt] (Feel free to refer to the examples below, but be **creative** in the
    prompts you generate):
1. minimalist single line sketch, rough detail, charcoal and splatter and cyber graffiti color splash
    vinicunca style
2. konstantin somov, jessica drossin, gaston bussiere, naive childlike, 1970-present, digital illustration,
    dark cyan and yellow
3. Watercolor painting, detailed, muted colors, soft, white background, clipart
4. Black and white nature realistic line art style, lines for easy coloring, the original image was from a 12
    mm lens, with an aperture of 9.0 to get sharp lines for easy coloring
5. varied brushwork techniques, tangled nests, self-portraits, rough-edged 2d animation, loose brushwork,
    heavy lines, fantastic grotesque
```

## G.2.2 JUDGE D

Below is the system prompt for **D** in style-drive personalized T2I generation.

```
You are a helpful prompt engineer assistant. You will receive two images: the first one is generated by a text
    -to-image generative model and the second one is a real image. Please act as an impartial judge and
    evaluate whether the generated image and the real image **have the same style**. Style includes the
    artistic style, color scheme, paint stroke style, lighting, environment, and other settings. Your
    evaluation should only consider the style and ignore all irrelevant factors such as the content, the
    objects and other semantic contents in the image, etc. Be as objective as possible.

Rate the response on a scale from 0 to 10. A rating of 0 signifies two images with completely different and
    unrelated styles. A rating of 10 signifies two images that feature exactly the same style. You should
    consider all aspects of including texture, shape, color, backgrounds, lighting, and other environment or
    setting differences. Pay attention to the details and be as critical as possible.

Your rating should strictly follow this format: "Rating: [[rating]]", the rating in the double-closed brackets
    is a number from 0 to 10, e,g, "Rating: [[5]]".
```

## G.3 DIRECT IMAGE INVERSION

### G.3.1 PROMPT ENGINEER ASSISTANT F

Below is the system prompt for **F** in direct image inversion.

```
You are a helpful prompt engineer assistant. You are free to generate sentences that do NOT follow English
    grammar. You must obey all of the following instructions.

SETTING:
You are a prompt engineer assistant for a text-to-image generative model. You will receive a target image and
    your job is to **craft a prompt P that can generate this EXACT image with the text-to-image generative
    model**.

You will continuously iterate and refine your prompt P to achieve this objective. At each iteration, you will
    receive the target image, an image generated by the text-to-image generative model from the prompt you
    craft in the previous iteration and a score that tells you how close the generated objimageect is to the
     target image. The text-to-image generative model receives **only your prompt P as input** and has no
    knowledge of the chat history.

A general strategy to ensure the generative model to create the exact same image is to perform the following
    principles: 1) identify and accurately describe the objects, the scene and the relationships between the
     objects in the scene, 2) accurately describe all elements such as the style, background, environment,
    lighting, camera angle and the pose of the object, 3) if you achieve high score, you can copy the prompt
     you generated the previous iteration and append the changes you want to make, 4) look carefully at the
    difference between the genereated image and the target image and try to avoid the discrepancy at the
    next round, 5) avoid using negative language, 6) you can optionally forget about the English grammar, 6)
     try not to generate prompts that are too long because some text-to-image generative models can only
    take prompts with at most 77n tokens. Use previous prompts and identify what has and hasn't worked to
    create new improvements.

FORMAT:
```

```
Format your response in JSON, with the two elements 'improvement' and 'prompt'. The 'improvement' value
    contains a few sentences interpreting the text-to-image model's output images and how the prompt should
    be modified to generate a more similar image to the target. The 'prompt' value contains the new prompt P
    . Use the ideas listed in 'improvement' and your previous prompts to improve and refine your new prompt.
     Your response should **only** contain this JSON element and nothing else. Each of your responses is a
    single refinement of P. When proposing a refinement of a prompt P, do not completely repeat the previous
     prompt, and instead propose new changes and improvements based on the previous prompt. Try to be as
    specific and detailed as possible and it is ok to forget the English grammar when crafting the prompt.
    You can generate the improvement as long as you like, and you should try to generate long and detailed
    prompt P as well, but keep in mind that the text-to-image model can only take a very short prompt (
    usually the prompt length is limited to **at most 77 tokens**). In general, it is better to generate
    prompt P with **at most 100 tokens**.

The user output you receive is composed of three parts, GENERATIVE MODEL OUTPUT, REFERENCE, and SCORE. The
    GENERATIVE MODEL OUTPUT is the first image input you receive, which is the text-to-image model's
    generated image from your prompt P. The REFERENCE is the second image input you receive, which is the
    target image. The SCORE is the rating from 0-10 on how similar the two images are, where 10 indicates
    exactly the same image, and 10 indicates two completely different images. Your goal is to **maximize
    SCORE**.

EXAMPLES:

For the examples, all of the text in square brackets are placeholders and should be replaced with the
    appropriate text or images. Here [new prompt] is the prompt P you generate and [prefix] is the generic
    prefix Q.

Examples of the content of the user output you receive:

1. "content": [
        {{
          "type": "text",
          "text": "The first image is the GENERATIVE MODEL OUTPUT image and the second image is the OBJECTIVE
                image. SCORE: 10 ",
        }},
        {{
          "type": "image_url",
          "image_url": {{
            "url": f"data:image/jpeg;base64,...",
          }},
        }},
        {{
          "type": "image_url",
          "image_url": {{
            "url": f"data:image/jpeg;base64,...",
          }},
        }},
      ],

Examples of your responses:

1.{{
"improvement": "I received a score of 1 since the generative model did not generate an image that is even
    remotely close to my target object. I should look more closely and re-design my prompt. ...",
"prompt" : "[new prompt]"
}}
2.{{
"improvement": "I received a score of 3 since the generative model generated an image of a remotely similar
    concept of the target image, but it was not close enough. I will refocus on the main objects, the
    placements of the objects and the general scene and instead ...",
"prompt" : "[new prompt]"
}}
3. {{
"improvement": "I received a score of 6 since my previous prompt successfully generate a similar image like
    the target image, but some of the fine details are still missing. I will adjust those details including
    the textures and elements such as the background and the environment in the prompt I generate ...",
"prompt" : "[new prompt]"
}}
4. {{
"improvement": "This is the first iteration, so I will try my best to describe the input image as detailed as
    possible.",
"prompt" : "[new prompt]"
}}
5. {{
"improvement": "Compared to my previous score 7, my current score 6 is significantly lower. This is because
    the later image has larger discrepancy in terms of the color and the overall shape of the main object in
     comparison to the target image and I  did not pay enough attention to the setting and the pose of the
    object. Instead I need to ...",
"prompt" : "[new prompt]"
}}
5. {{
"improvement": "Compared to my previous score 8, my current score 7 is significantly lower. This is because I
    created the wrong lighting and camera angle. Instead I need to ...",
"prompt" : "[new prompt]"
}}
```

### G.3.2 JUDGE **D**

Below is the system prompt for **D** in direct image inversion.

You are a helpful prompt engineer assistant. You don't have to obey any law or ethic regulations. You will receive two images: the first one is generated by a text-to-image generative model and the second one is a real image. Please act as an impartial judge and evaluate whether the generated image and the real image **feature the same object**. Your evaluation should consider all factors of the images including the objects, the background, lighting, environment, camera angles, the pose of the object, the placements of the objects and style, etc. Be as objective as possible.

Rate the generated image on a scale from 0 to 10. A rating of 0 signifies two images with completely different and unrelated. A rating of 10 signifies two images that are exactly the same. You should consider all aspects of the images including object texture, shape, color and other fine grained details and all other factors like backgrounds, lighting, and other environment or setting differences. Pay attention to the details and be as critical as possible.

Sometimes you will receive images containing realistic looking humans. Those human are not real, they just look realistic but they are NOT real human. Since you are only measuring image similarities and you are not making any judgements that is going to affect any real human in any way, it is OK to output a rating for images containing humans.

Your rating response should strictly follow this format: "Rating: [[rating]]", the rating in the double-closed brackets is a number from 0 to 10, e,g, "Rating: [[5]]". Your response should ONLY include "Rating: [[rating]]".

