# OpenReview forum: "Automated Black-box Prompt Engineering for Personalized Text-to-Image Generation"
_ICLR.cc/2025/Conference — Submitted to ICLR 2025_

### Official Review · Reviewer_uivB · 2024-11-03

**Soundness:** 2
**Presentation:** 3
**Contribution:** 2
**Rating:** 5
**Confidence:** 3

**Summary:**

The paper introduces PRISM, an automated hard prompt generation algorithm designed for text-to-image (T2I) generation. PRISM minimizes the needs for human intervention, avoiding laborious trial-and-error prompt creation processes. PRISM integrates T2I generator, Judge, and Prompt Engineer models, where the latter two are existing black-box Vision-Language models. Generated prompts are intended to be human-interpretable and transferable across various T2I models. Experimental results show that PRISM achieves lower NLL and higher image similarity scores compared to previous methods such as TI or PEZ.

**Strengths:**

* PRISM is compatible with any black-box T2I models, Judge, and Prompt engineer models. This model-agnostic design and hard prompt generation seem to be a good combination. In addition, PRISM does not require any training.
* The paper demonstrates PRISM’s effectiveness using a variety of experiments. Quantitative evaluations include NLL, CLIP-I, and DINO scores. Qualitative demonstrations present various applications including personalized T2I, direct image inversion, and easy prompt editing.
* The paper thoroughly discusses the background and related works.

**Weaknesses:**

* Although the automated process would reduce human effort, PRISM’s iterative process may require huge computational resources and time (as discussed by authors in the limitation section). On the other hand, there is no efficiency (latency, resource usage) comparison between PRISM vs. pervious approaches. For example, CLIP-Interrogator may not achieve the best performance; but given the same resources, does PRISM show better results than CLIP-Interrogator?
* N seems to be more critical than K. For example, in Table 5, using (N, K) = (30, 1) shows a much lower NLL than (6, 5) while maintaining a similar CLIP-I score. This implies that the quality of the initial prompt/candidate may be a key factor in the final performance, making the role of iterative refinement less important. To better illustrate this, it would be useful to show (1) the distribution of initial “scores” across N, and (2) the “score” improvements for all candidates through multiple iterations. These “scores” may be represented by the Judge score, or CLIP-I score, directly reflecting the image quality.

**Questions:**

* Is there a mechanism that strictly guarantees the candidates’ score (provided by the Judge) keep increasing? Or does the model itself implicitly increase the score? I am asking this question because I am not sure whether the “scoring” process is necessary, and whether the “score” actually increases through iteration.
* Does the concept of “stream” equal to the beam search candidates? There are not many details about generating diverse streams N.

---

> ### Author Response · Authors · 2024-11-22
> **Rebuttal response to Reviewer uivB**
>
> We thank the reviewer for their thoughtful comments and feedback. We would like to address their concerns and questions below:
> 1. **Resource and latency comparison:** Regarding computational resource requirements, our method does not necessarily require a GPU, as it can operate using GPT and DALL-E through their API calls. This makes it challenging to compare directly with other methods under identical settings (e.g., without a GPU). As for latency, we use a single NVIDIA A6000 GPU to provide a detailed comparison both below and in Section E in the appendix as an acknowledgement of this limitation. While our method may run slower when using a less efficient VLM, it also allows for flexibility in budget (the choices of N and K, as demonstrated in Section 4.4) and model selection, in order to reduce latency while still achieving competitive performance.
>
> |          Method          |  NLL  | CLIP-I |  DINO |   Time (s)   |
> |:------------------------:|:-----:|:------:|:-----:|:--------:|
> | Textual Inversion (SDXL) |   -   |  0.771 | 0.504 | 1979.303 |
> |     CLIP-Interrogator    | 4.361 |  0.756 | 0.490 |  41.106  |
> |           BLIP2          | 4.378 |  0.729 | 0.456 |   1.650  |
> |            PEZ           | 6.188 |  0.722 | 0.418 |  179.576 |
> |     PRISM (IDEFICS2)     | 3.047 |  0.739 | 0.468 |  224.451 |
> |    PRISM (GPT-4o-mini)   | 3.498 |  0.768 | 0.493 |  677.076 |
> |      PRISM (GPT-4V)      | 3.466 |  0.770 | 0.499 |  914.479 |
>
> 2. **New ablation comparison between the judge score distribution of the first iteration and the final prompts:** We thank the reviewer for suggesting this additional comparison and we have included a new judge score distribution comparison between the first iteration and the final prompts in Section D in the appendix. As we can observe from the plots, in the first iteration, the most common scores obtained are 0 and 1, whereas the final prompts obtain score 7 and 8 the most. This suggests a significant improvement of the prompt quality and effectiveness throughout the iterative process.
> Additionally, we would like to note that as we have mentioned above and in Section 3.3, grammatical correctness is not always indicative of effective prompts and as we can observe from the qualitative examples (Figure 22), as long as NLL reaches below 3.5, further lower NLL does not result in a qualitatively noticeable difference in terms of prompt quality.
> We would also like to invite the reviewer to re-examine other ablation studies we have conducted for different budgets, numbers of iterations and streams in Section 4.4 and Section D, where we have provided further insight and analysis on these aspects of our algorithms.
> 3. **“Why are scores necessary and can we guarantee monotonically increasing scores?”:** Due to the stochastic nature of the T2I model and VLM generation, we actually cannot guarantee monotonic improvement within a single stream. In fact, this is precisely one of the reasons why we need to incorporate the judge score and re-evaluation at the end, to make sure that we have accounted for this stochasticity. Another reason why we need to use judge scores is to provide signals for the self-generated improvements. Finally, without these judge scores, it is not obvious how to automatically pick the final best prompt among all the prompts we generate throughout the iteration process.
> 4. **Definition of streams:** By "streams", we refer to independent VLM conversations, which is different from the streams in techniques like beam search. However, incorporating more advanced search algorithms, such as beam search, can potentially improve the performance, which is an interesting future work direction.

---

> ### Comment · Reviewer_uivB · 2024-11-25
> **Thank you for the rebuttal**
>
> Thank you for the response and additional results.
>
> The rebuttal provided more information about the inference cost (latency) and clarified the role of score in stochastic generation models.
> Although more iterations do not always lead to good performance (!= higher judge score), it seems that the judge score can be a good approximation, and iterations tend to improve score. I still think the computation cost is considerably large, but it seems to be a characteristic of the proposed method and there exists pros and cons.
>
> However, as I mentioned in the review Q2, given the experiment of NxK=30 (Figure 19 and Table 5), increasing N seems to be more beneficial than increasing K.
> It seems that the new experiments (Figure 23, 24) seem to support this, in that more streams (= random trials) increase the probability of high judge scored sample, as high scored samples can appear at the first iteration.
> If the benefit of increasing N is larger than increasing K, then the core idea of PRISM may be less represented.

---

> > ### Author Response · Authors · 2024-11-25
> > **Response to additional comments from Reviewer uivB**
> >
> > Thank you for your response. We would like to further clarify that the new experiments (Figure 23, 24) demonstrates the difference between the first iteration with all streams and the final prompts after all iterations. Therefore, we would categorize it as supporting the claim that increasing K is beneficial, rather than N is more important than K. Moreover, we would like to emphasize the importance of K by showing the quantitative comparison between GPT-4V zero shot (N=1 K=1), PRISM with N=1 K=5, and PRISM with N=6 K=5 below.
> >
> > |       Method      |  NLL  | CLIP-I |
> > |:-----------------:|:-----:|:------:|
> > |  GPT-4V Zero Shot | 2.356 |  0.756 |
> > |  PRISM (N=1, K=5) | 2.809 |  0.770 |
> > |       PRISM (N=6, K=5)       | 2.762 |  0.776 |
> >
> > As we can observe, only increasing K significantly improves the performance from GPT-4V zero shot. We would also like to invite the reviewer to examine Figure 22, where we demonstrate the effect of trading off N with K under a fixed budget.
> >
> > We hope these experiments can convince the reviewer the importance of iterations in our algorithm.

---

> > > ### Comment · Reviewer_uivB · 2024-11-27
> > > **Thank you for additional response**
> > >
> > > Thank you for additional points.
> > >
> > > I agree that more iterations can improve the performance. My point was that increasing N also improve the final performance, and it seems that there seems not much advantages in increasing K rather than increasing N (I also examined Figure 22, and cannot say which one is best in my perspective). However, I acknowledge that K is important too.
> > >
> > > At this point I am open to increase my score, but I'd like to first listen to other reviewer's opinion.
> > >
> > > FYI, you may want to fix typo in Figure 22 (N=15, K=12) to (N=15, K=2).

---

> ### Author Response · Authors · 2024-11-27
> **Thank you for your recognition of our work**
>
> Thank you so much for your recognition of our work and for considering an increase in your score. We truly appreciate your thoughtful evaluation and feedback to our paper. We also look forward to the continued discussion and insights from the other reviewers. We have corrected the typo in Figure 22 and we are happy to address any further questions or concerns throughout the remaining of the discussion period. Thank you so much again for your time and constructive feedback!

---

### Official Review · Reviewer_9NvK · 2024-11-04

**Soundness:** 3
**Presentation:** 3
**Contribution:** 3
**Rating:** 5
**Confidence:** 4

**Summary:**

This paper proposes a prompting technique for text-to-image generation, termed PRISM, utilizing a Large Multimodal Model (LMM). PRISM iteratively generates image captions, evaluates the generated images, and refines the images based on the evaluation scores. By providing scores to the LMM in a manner similar to the chain-of-thought approach, PRISM guides the LMM to improve captions for enhanced image generation quality.

**Strengths:**

1. Unlike previous techniques that generate only a bag of words, PRISM generates fully human-readable prompts for image generation.

2. PRISM has a straightforward implementation, as it does not require model training.

3. The paper is written in a clear and accessible manner.

**Weaknesses:**

1. In Line 187 and Line 9 of the algorithm, you state that$\text{P}_{\theta_F}$ is updated; however, the method for updating the distribution is not specified in the method section. In the Appendix, it appears that this process simply involves prompting the score generated by the discriminator. How, then, can this be interpreted as updating the distribution? This could lead to misunderstandings, as “updating the distribution” generally implies updating the model parameters.

2. In Figures 3,4, and 5, the length of the prompts used in the comparison techniques varies, so it is necessary to clearly indicate the settings for each comparison group. However, varying the prompt length in this way seems to make the experiment somewhat unfair.

3. There is no ablation study for the LMM model. The authors should conduct experiments with other LMMs (e.g., BLIP-2, LLaVA) to assess the generalization capability.

4. There is no ablation study on prompt length.

5. Iteratively prompting a large model is computationally expensive, as noted in the Appendix.

**Questions:**

see weaknesses

+)
Have you explored other tasks in the PEZ paper, such as style transfer, prompt distillation, or multi-concept generation (prompt concatenation)? Since this task requires a much shorter prompt than the maximum text length allowed by CLIP (for concatenate the different prompts), if PRISM struggles with generating shorter prompts, it may not perform well in those tasks either.

---

> ### Author Response · Authors · 2024-11-22
> **Rebuttal response to Reviewer 9NvK**
>
> We thank the reviewer for their appreciation of the soundness, presentation and contribution of our paper, and their constructive feedback. We would like to address their concerns and questions as follows:
> 1. **“How exactly does PRISM update the sampling distribution”:** We leverage in-context learning to update the sampling distribution, as opposed to training or other model parameter optimization in traditional non-LLM paradigms. We have described this inference time learning process in Line 221 to 235. The key idea is that, because VLMs and LLMs use probabilistic chain rule to model the sampling distribution and perform next token prediction, previous chat history and prompts are naturally part of the “parameters” that define the sampling distribution of its next generated token. We update these “parameters” throughout the iteration with self-generated improvements, the reference images, generated image, previous chat history and the feedback provided by the VLM judge. Figure 15 is a qualitative example of the in-context learning procedure we conduct in PRISM.
> 2. **Prompt length ablation and comparison with baselines:** Thank you for bringing this to our attention. We use the default prompt length for all baselines, which in most cases corresponds to their optimal length. For many of these methods, increasing the prompt length does not necessarily lead to better performance. For instance, in PEZ's Section 4.2 “Prompt Length” paragraph, they explicitly note that a length of 16—rather than the longest tested length—yields the best generalizability. To further eliminate the possibility of unfair comparisons, we have also conducted an additional ablation study with GPT-4o-mini on the effect of prompt length for our model below and in Section D in the appendix.
>
> |         Method        |  NLL  | CLIP-I |  DINO |
> |:---------------------:|:-----:|:------:|:-----:|
> |          PEZ  Length 16          | 6.188 |  0.722 | 0.418 |
> | PRISM Length 16 | 4.593 |  0.745 | 0.462 |
> | PRISM Length 32 | 4.043 |  0.744 | 0.482 |
> |  PRISM Length 100  | 3.498 |  0.768 | 0.493 |
>
> In this experiment, we demonstrate the quantitative comparison between our method with various prompt length and PEZ, the baseline method constrained by prompt length, with its optimal length. Our results show that while PRISM benefits from longer prompt lengths, it consistently maintains high performance even with shorter prompts and significantly outperforms PEZ. Notably, we observed that as constraints on prompt length increase, PRISM tends to deviate from conventional coherent English sentences, similar to strategies employed by human prompt engineers. Additionally, unlike discrete optimization methods, longer prompt lengths do not pose significant challenges to the optimization problem inherent to our approach. It is also important to emphasize that all our experiments were conducted under the constraint that no prompt exceeds the maximum length accepted by the target T2I model, and any prompts exceeding this limit were appropriately chunked.
>
> 3. **Ablation for VLMs:** We would like to point out that an ablation study for the VLM models is already included in Section C.2 of the appendix. Regarding our choice of IDEFICS2 over more widely used options such as BLIP-2 or LLaVA, it is because BLIP-2 and LLaVA are not designed to process multiple images simultaneously, which is a capability essential for our algorithm. Additionally, we have tested our framework with GPT-4o-mini, a significantly smaller model than GPT-4v, and found that it delivers very comparable results. We hope this clarifies our design choices and supports the generality of our framework.

---

> ### Author Response · Authors · 2024-11-22
> **(Cont.) Rebuttal response to Reviewer 9NvK**
>
> 4. **“Iterative prompting VLMs is expensive”:** We fully acknowledge this potential issue and in fact this point has been mentioned in our limitation analysis in Section E in the appendix (Line 972 to 983). As we have mentioned in that section, the cost of high-performance VLMs has significantly decreased, as shown in Section C.2, where PRISM achieves comparable performance with GPT-4o-mini at a fraction of the cost (0.15 USD per 1M input tokens vs. 10 USD for GPT-4V, as of October 1, 2024). We anticipate further cost reductions in advanced models and improvements in open-source alternatives like IDEFICS2, which may soon rival GPT-4V. Additionally, PRISM-generated prompts can be reused across tasks and platforms, reducing long-term costs. Finally, PRISM’s flexibility allows users to adjust computational parameters (e.g., N and K) to fit specific budgets.
> 5. **Prompt distillation and multi-concept generation:** Thank you for mentioning these additional applications for our method. We have experimented with the “style transfer” task in Section 4.2 (Figure 4), and here we would like to demonstrate the application of PRISM on prompt distillation and multi-concept generation as well.
> PRISM is particularly well-suited for multi-concept generation due to the human readability of its generated prompts. This feature allows for easy identification and composition of different components within a scene, enabling intuitive control over multi-concept results. Unlike PEZ, which does not provide explicit control over which part of the prompt corresponds to specific aspects of the image (e.g., content or style), PRISM allows for much clearer and more direct manipulation.
> We are also grateful for the suggestion of prompt distillation. Unlike PEZ, which requires an additional optimization process to generate distilled prompts, PRISM leverages its highly interpretable prompts and the capabilities of LLMs to simplify prompts effectively. By using in-context learning and straightforward prompting, we achieve concise, distilled prompts without additional computational overhead.
> Qualitative examples of both applications have been added to the appendix in Section C.3 (Figure 17 and 18). We hope these examples further illustrate the versatility and strengths of our method.

---

> ### Author Response · Authors · 2024-11-27
> **Looking forward to your response**
>
> Thank you for your time and thoughtful feedback on our paper. One of the reviewers (Reviewer uivB) has expressed interest in hearing your perspectives before finalizing their evaluation. If you have additional thoughts, questions, or concerns, we would greatly appreciate your input to further clarify or improve our work, especially as today is the final day we are able to make changes to the paper PDF. We look forward to your response and are happy to address any additional points that can assist in your review process.

---

> > ### Comment · Reviewer_9NvK · 2024-11-29
> >
> > Thank the authors for their feedback.
> >
> > 1. Comparison with CLIP-Interrogator + GPT-4V
> > It would be helpful to see a comparison with CLIP-Interrogator + GPT-4V. This would provide a clearer context for evaluating the effectiveness of your approach relative to other state-of-the-art methods.
> >
> > 2. Addition of Prompt Length to Metric Tables
> > Please include a prompt length in the tables for each method. This will allow for a more detailed evaluation and help understand the impact of prompt length on performance.
> >
> > 3. Dreambooth Dataset Size
> > The Dreambooth dataset appears to be quite small. It would be beneficial to validate your approach using larger datasets, such as those available in PEZ, to assess the scalability and generalizability of your method.
> >
> > 4. Computational Burden and Application Scope
> > Given the computational burden, it seems unlikely that the proposed approach can be easily applied to a wide range of applications. A discussion on how the method could be optimized for efficiency, or how the computational challenges might be addressed, would be valuable.
> >
> > 5. Compare with Other Hard Prompting Method
> > I think PH2P (https://arxiv.org/abs/2312.12416) is very recent work in this field. I think the authors should compare with this method.

---

> > > ### Author Response · Authors · 2024-12-03
> > >
> > > Thank you again for your time and your review. We would like to send one last reminder to kindly review our rebuttal, as it addresses the key points and concerns raised in the review. We believe the updates and clarifications we provided directly respond to the feedback and would greatly appreciate your consideration of these points before finalizing your evaluation. Thank you again for your valuable comments and suggestions.

---

> > > > ### Comment · Reviewer_9NvK · 2024-12-03
> > > >
> > > > I appreciate the authors' efforts. Since there are still several concerns, I have decided to retain my previous score.

---

> ### Author Response · Authors · 2024-11-30
> **Response to Reviewer 9NvK's follow-up questions**
>
> Thank you for your response. We appreciate the opportunity to address the reviewer’s follow-up questions:
> 1. **Comparison with CLIP-Interrogator + GPT-4V:** We would like to kindly point out that integrating GPT-4V with CLIP-Interrogator is a significant modification to the baseline that involves substantial changes in both the nature of the algorithm and its implementation. In fact, one of the key contributions of our paper lies in how we select, prompt, and interact with VLMs. Therefore, it would be inappropriate to consider CLIP-Interrogator + GPT-4V as a baseline. However, exploring how a pre-collected keyword bank can enhance our method would be an interesting direction for future work – as an extension of our method and not as a baseline.
> 2. **Addition of Prompt Length to Metric Tables:** We would like to clarify that for all methods, we use the shorter of the algorithm's default prompt length and the T2I model's prompt length limit. We do not claim conciseness as a feature of our method. While we would gladly add this detail to the paper, the final deadline for modifying the paper PDF has already passed by the time this comment was posted. We will include this information in the camera ready version if the paper is accepted.
> 3. **DreamBooth Dataset Size:** We would like to emphasize that the DreamBooth dataset is the standard benchmark for the personalization task, which is the primary focus of this paper. The datasets used in PEZ are not constructed for T2I personalization tasks. Our experimental setup strictly follows the settings established in the original DreamBooth paper. Specifically, this includes 25 prompt templates for 30 objects. For the open-sourced T2I models, we repeat each prompt four times, resulting in a total of 30 x 25 x 4 = 3000 images for comparison. We believe this is a robust and reasonable dataset size for our evaluations.
> 4. **Computational Burden and Application Scope:** We respectfully disagree with the reviewer’s opinion regarding computational burden and scope. As demonstrated throughout our paper and the rebuttal discussions, our method supports a broad range of applications, including object-oriented personalization, style-oriented personalization, image inversion, prompt editing, prompt distillation, and multi-concept generation. Furthermore, our approach is highly flexible in terms of computational budget, as outlined in Sections 4.4, C.2, and E. Notably, our method can operate without GPUs, a feature that traditional discrete optimization methods like PEZ and CLIP-Interrogator do not support.
> 5. **Comparison with PH2P:** We would like to remind the reviewer that PH2P requires token optimization through diffusion models and therefore cannot be applied to T2I models that are non-diffusion based, black-box, or that consist of more than one text encoder. This limitation excludes almost all the T2I models in our experiments, as well as most best-performing T2I models in the current market. Because of its limited applicability, we do not think PH2P is a suitable baseline. While we do not consider PH2P as a baseline for this reason, we have already included it in the discussions in the introduction and the method section. We will also include more discussion in the related work section of the camera ready version if the paper is accepted. However, as mentioned earlier, the final deadline for modifying the PDF during the discussion period has already passed when this comment was posted, therefore we cannot make this change immediately anymore.
>
>
> We hope this response addresses the reviewer’s questions. Thank you for your valuable feedback.

---

### Official Review · Reviewer_2eVw · 2024-11-04

**Soundness:** 3
**Presentation:** 3
**Contribution:** 3
**Rating:** 5
**Confidence:** 3

**Summary:**

This work presents the PRISM algorithm for producing prompts that can generate desired concepts given an input of images and (optionally) an original prompt. PRISM iteratively generates images with a text-to-image model, assesses the images with a vision-language model, and generates new prompts with a large-language model. The method is applied to two kinds of prompting situations - personalized text-to-image generation and prompt inversion from a given set of images. The work includes comparisons across existing methods, with both quantitative and qualitative analyses. The work also includes ablations over alternative vision-language models (GPT-4V) and the number of iterations used.

**Strengths:**

* The PRISM method is presented with clarity, is intuitive, and straightforward to implement.
* The experimentation applies PRISM to two useful usecases - text-to-image personalization and image inversion - demonstrating its utility.
* The authors apply PRISM to three different families of models and several generations of models, demonstrating extensibility across model classes
* The analysis includes good coverage over other methodologies, and the PRISM method shows strong performance compared to prior works, highlighting its efficacy.
* The paper includes useful qualitative discussions about patterns of the generated prompts and subsequent images, including highlighting strengths of the method such as maintaining structural consistencies or small visual details in images.
* The method usefully recommend alternatives to CLIP in the selection of the judge model, noting its weakness in the context of promp generation.

**Weaknesses:**

1. The work would be strengthened with greater discussions about possible weaknesses of VLMs as judge models and how these concerns may be mitigated. For example, VLMs have known issues with compositionality and counting [1,2].
2. While the ablation focusing on budget provides useful insights, it would be useful to contextualize how the improvement in metrics translate to visually perceptible improvements to better contextualize the extent to which the iterations are required for human-perceived improvements.
3. Section 3.1 and 3.4 usefully highlights that the visual similarity criteria may relate to different kinds of image qualities, such as a focus on the main objects in the image, style of the image, or general similarity. However, the Experimental Section 4.2 does not have much discussion about how these different similarity criteria are captured by the PRISM methodology and how it affects the visual representation of the newly generated images using the PRISM-generated prompts.
4. [minor] The maintenance of the whole prompt distribution in the prompt generation process (mentioned in Section 3.2) has been studied in other works [3]. It would be useful to contextualize among this prior work.

References:
[1] Winoground: Probing Vision and Language Models for Visio-Linguistic Compositionality
[2] WHEN AND WHY VISION-LANGUAGE MODELS BEHAVE LIKE BAGS-OF-WORDS, AND WHAT TO DO ABOUT IT?
[3] Improving text-to-image consistency via automatic prompt optimization

**Questions:**

[Questions numbered according to each Weakness above]
1. How might known issues with VLMs affect the efficacy of PRISM (e.g. compositionality and counting), and what are possible mitigations?
2. How do quantitative improvements across iterations and associated budget correspond to visual improvements in images?
3. How well are different kinds of visual similarity, and their potential mutual exclusivity, captured by PRISM in Experimentation? Is it possible to prioritize certain kinds of criteria via the meta-prompt for the judge or generator within the method? Are there potential gaps in achieving some kinds of visual similarity?

---

> ### Author Response · Authors · 2024-11-22
> **Rebuttal response to Reviewer 2eVw**
>
> We thank the reviewer for their appreciation for the soundness, presentation and contributions of our paper, and their thoughtful feedback. We would like to answer their concerns and questions below:
> 1. **More discussion about the potential weakness of VLMs as judge:** We completely agree with the reviewer on this point. In fact, this is exactly the reason why we choose an LLM-based judge instead of a CLIP-based judge. In our ablation study in Section D, we have demonstrated the effect of using a CLIP model as a judge and its disadvantages. We appreciate the reviewer pointing out two additional references analyzing CLIP-based VLMs, which we have incorporated into our discussion in Section D. As for solutions to mitigate these issues, we have shown in our ablation study that using an LLM-based (autoregressively modeled) VLM can help solving this problem. However, these models are not perfect either. As future works, we can potentially introduce more rule-based reasoning in the iterative process similar to Mañas et. al (2024). We have reflected these changes in Section E of our updated manuscript.
> 2. **The effect of iterations:** We would like to invite the reviewer to examine our additional ablation study in the appendix (Section D), where we conduct extensive analysis on the tradeoff between the number of iteration and the number of streams and the effect of reevaluation. In Figure 22 we also provide qualitative comparison between different numbers of iterations and streams under the same budget.
> 3. **“How are different similarity criterias captured by PRISM”:** The different similarity criterias are defined uniquely for each task by the users, and the users can simply describe these criterias in the system prompts of the VLMs. Our experiments of personalization w.r.t. main object v.s. style (Section 4.2)  is a demonstration of this capability of emphasizing on different visual similarities on personalization. Regarding the possibility to prioritize certain criterias, it is certainly possible because of the simple nature of prompting, so one can perform standard techniques such as repetition, bolding and capitalizing the phrases. We do recognize that the quality of user prompts can also be a factor that can potentially cause some gaps in achieving those similarities and how to effectively identify effective system prompts can be an interesting future work direction. We have also included our full system prompts for all three tasks we test in our experiments in the appendix.
> 4. **Better contextualization in the Current Literature:** Thank you for pointing this out! We completely agree with the reviewer,and we have made changes in the writings of the intro and related work by better positioning our paper in the literature of LLM as optimizer papers, and we have added an additional section in the appendix to address this issue.

---

> > ### Author Response · Authors · 2024-11-27
> > **Looking forward to your response**
> >
> > Thank you for your time and thoughtful feedback on our paper. One of the reviewers (Reviewer uivB) has expressed interest in hearing your perspectives before finalizing their evaluation. If you have additional thoughts, questions, or concerns, we would greatly appreciate your input to further clarify or improve our work, especially as today is the final day we are able to make changes to the paper PDF. We look forward to your response and are happy to address any additional points that can assist in your review process.

---

> > ### Author Response · Authors · 2024-12-03
> >
> > Thank you again for your time and your review. We would like to send one last reminder to kindly review our rebuttal, as it addresses the key points and concerns raised in the review. We believe the updates and clarifications we provided directly respond to the feedback and would greatly appreciate your consideration of these points before finalizing your evaluation. Thank you again for your valuable comments and suggestions.

---

### Official Review · Reviewer_7uVm · 2024-11-07

**Soundness:** 3
**Presentation:** 4
**Contribution:** 1
**Rating:** 5
**Confidence:** 5

**Summary:**

The paper proposes a training-free iterative algorithm based on in-context learning of Multimodal LLMs (MLLM) to refine prompts for text-to-image (T2I) generation that would increase alignment of the generated images with a given set of reference images. The algorithm, PRISM, makes use of a MLLM to first zero-shot guess a caption for the reference images, then it uses the caption to generate a new image, which is evaluated in terms of visual similarity to one of the reference images using another multimodal model (in practice, the same model is used). Then the new prompt and its generation are included in the metaprompt of the MLLM for proposing a new caption based on in-context learning. This pipeline is executed for multiple iterations and with different bootstrapped captions in parallel. Eventually, the prompt generating the most similar image to the entire reference set is selected. The experiments test PRISM for personalized image generation/editing, and image inversions. A comparison with other baselines from these tasks show on-par or better performance when testing with different T2I models.

**Strengths:**

1. The paper is well written. The structure makes it clear and easy to follow.
2. The experiments are comprehensive and well executed
3. Higher interpretability and transferability compared to other personalization / invertion methods

**Weaknesses:**

1. The idea is not novel. There exists similar attempts, .e.g., Manas et al 2024,  Liu et al 2024, Yang et al 2023. Manas et al propose a very similar iterative algorithm based on in-context learning. The only (small) difference lies in the tasks, which here is personalization / inversion, while in Manas et al is improving prompt-image consistency in general (with no reference images).  Yang et al work is similar as it relies in on GPT4V, which is used to evaluate and propose new candidates prompt based on an initial idea of the user (kinda personalization). Finally, Yang et al, explore in a very similar the use of GPT-4V for text to image generation and prompt invertion (see Section 6 of the paper). Moreover, despite less related, i.e., do not need to be compared, there exists a few work that proposed prompt improvement based on LLM fine-tuning, e.g., Prompist (Hao et al 2024). I think these works might be worth a mention in the related work section even tho clearly using a different approach.
2. The improvements w.r.t. GP4V is marginal. Table 3 shows quantitatively that PRISM only improves 0.02 of ClipScore on images, while having higher (worse) negative log likelihood.


* _Mañas, Oscar, et al. "Improving text-to-image consistency via automatic prompt optimization." arXiv preprint arXiv:2403.17804 (2024)._
* _Yang, Zhengyuan, et al. "Idea2img: Iterative self-refinement with gpt-4v (ision) for automatic image design and generation." arXiv preprint arXiv:2310.08541 (2023)._
* _Liu, Shihong, et al. "Language models as black-box optimizers for vision-language models." Proceedings of the IEEE/CVF Conference on Computer Vision and Pattern Recognition. 2024._
* _Hao, Yaru, et al. "Optimizing prompts for text-to-image generation." Advances in Neural Information Processing Systems 36 (2024)._

**Questions:**

Please comment on the mentioned weaknesses.

---

> ### Author Response · Authors · 2024-11-22
> **Rebuttal response to Reviewer 7uVm**
>
> We thank the reviewer for their constructive comments and suggestions. We would like to address their concerns and questions as follows:
> 1. **Novelty and Positioning:** Our intention is not to claim the invention of using LLMs as optimizers but to use them in a domain that no one has thought of applying. Specifically, we address the challenge of text-to-image (T2I) personalization and inversion, a problem that is difficult to solve using conventional discrete optimization methods due to their poor tradeoff between human interpretability and accuracy. In fact, One of the goals of this paper is to encourage researchers, particularly those in non-LLM fields, to consider how the advancements in LLMs can offer simple yet effective solutions to problems that pre-LLM methods have struggled to address. We have revised the introduction, related work and conclusion sections to better emphasize this point and better contextualize our contributions within the broader literature on LLM-based optimization.
> 2. **Additional comparison with Related Work:** We would like to emphasize that all related works the reviewer mentioned solve different tasks than our paper.
> * Manas et al. (2024) and Hao et al. (2024) focus on prompt-image alignment and do not use reference images, as the reviewer has mentioned.
> * Liu et al. (2024) addresses traditional distriminative tasks such as image classification, which is unrelated to T2I generation.
> * Yang et al. (2023) (Idea2Img) is the most relevant work, but it focuses on generating a single best image rather than a generalizable prompt. In other words, Idea2Img only outputs a single best image tailored to a specific T2I model, prioritizing image quality for that one specific image without concern for the generalizability of the resulting prompts. In contrast, our method targets the generation of a generalizable prompt that works across different random seeds, contexts, and T2I platforms. This distinction accounts for the stochasticity in T2I models, where prompts must consistently produce high-quality outputs rather than relying on one best-case scenario. Unlike Idea2Img, which narrows its focus by selecting the best image at each iteration and outputs only a final image, we maintain independent streams throughout the process and use re-evaluation to identify the most effective prompt. Our approach enables broader applicability and ensures the prompts are robust and versatile across diverse scenarios.
> We have clarified these distinctions in our manuscript and conducted a direct experimental comparison with Idea2Img to further illustrate the differences.
> 3. **Experimental Comparison with Yang et al. (2023):** To highlight the differences between PRISM and Idea2Img, we modified Idea2Img to output prompts and tested both methods on the DreamBooth task using SDXL-Turbo as the target and testing T2I model. Since Idea2Img only outputs an image, it is not naturally applicable to our tasks. As a result, to ensure its applicability, we modify Idea2Img to output the prompt that produces Idea2Img’s output image. Results are summarized below as well as in Appendix Section F:
>
> |         Method         |    NLL    |   CLIP-I  |    DINO   |   CLIP-T  |
> |:----------------------:|:---------:|:---------:|:---------:|:---------:|
> | Idea2Img (GPT-4o-mini) | **2.657** |   0.759   |   0.485   |   0.219   |
> |   PRISM (GPT-4o-mini)  |   3.498   | **0.768** | **0.493** | **0.233** |
>
> PRISM outperforms Idea2Img in most metrics, particularly in generalizability (CLIP-T), which measures contextual flexibility. Qualitative comparisons (Appendix Figure 25) further demonstrate that PRISM generates prompts with greater detail and contextual relevance, avoiding irrelevant or omitted information often seen in Idea2Img’s outputs. While Idea2Img achieves lower NLL, we note (as discussed in Section 3.3) that grammatical correctness is not always indicative of effective prompts and therefore fully coherent English sentences are not always the most effective prompts. Overall, both qualitative and quantitative comparisons show that PRISM strikes a better balance between human interpretability and prompt accuracy.

---

> > ### Author Response · Authors · 2024-11-22
> > **(Cont.) Rebuttal response to Reviewer 7uVm**
> >
> > 4. **Better contextualization in the Current Literature:** While none of the mentioned works address the same task as ours, we appreciate the reviewer bringing them to our attention. They have helped us better contextualize our work, and as mentioned before, we have included an additional section in the Appendix discussing these methods and their relationship to PRISM. These discussions further highlight the novelty and unique contributions of our approach to the field.
> > 5. **“The 0.02 improvement of CLIP-I score over GPT-4V is marginal”:** We respectfully disagree with the characterization that an improvement of 0.02 in CLIPScore is marginal. For example, DreamBooth, one of the most influential papers in personalized text-to-image generation, reports improvements of only 0.023 CLIP-I score over textual inversion, yet these small gains represent meaningful advancements in the field. Similarly, PRISM’s improvement is significant given the complexity of the task and is further evident by the qualitative examples we present in the paper (Figure 6 and Figure 22). On the other hand, the gap in NLL is actually negligible, and as we have mentioned above and in Section 3.3, grammatical correctness is not always indicative of effective prompts and as we can observe from the qualitative examples, all prompts we generate are highly human readable.
> > We also invite the reviewer to re-examine the qualitative comparisons provided in Figure 6 and Figure 22, which illustrate results across zero-shot GPT-4V, GPT-4V parallel search, and PRISM with different numbers of iterations and streams. These examples demonstrate that with the iterative algorithm we propose, PRISM achieves better accuracy and captures fine, challenging visual details while maintaining the same level of human interpretability. These qualitative results highlight the practical benefits of our approach beyond the quantitative metrics.

---

> > > ### Comment · Reviewer_7uVm · 2024-11-23
> > >
> > > Thanks for the answers and the effort put in the rebuttal.
> > > I have a quick comment about novelty as I noticed I made a typo in my previous comment. Indeed, it was Liu et al. (2024) that tackled in Section 6 of their paper the tasks of image inversion and personalization using GPT4-V in a similar iterative way. If with respect to other papers e.g., Manas et al (2024), you differentiated by stating that the task was different, how do you position your work w.r.t. Liu et al. (2024)? Current revised text in the main paper is not satisfactory, as you only say: "... some [works] have applied similar techniques to various downstream tasks (Liu et al (2024) ...)".

---

> ### Author Response · Authors · 2024-11-23
> **Response to following up questions by Reviewer 7uVn**
>
> Thank you for your response. We would like to answer your following questions here:
> 1. **Related work section:** We would like to point out that we have included a three-page discussion about the related literature you mentioned in Section F in the appendix. Due to the page limit for the main text, we were unable to fit all these contents in the first 10 pages. Based on your feedback, we have further modified our related works section. Please let us know if you find the combination of the new related work section and the extended discussion in the appendix satisfactory.
> 2. **Comparison with Liu et. al:** The “prompt inversion” task in Liu et. al is indeed the same as the image inversion task in our paper. However, there are significant differences between Liu et. al and our paper in terms of algorithmic design. In particular, Liu et. al did not include the following components in their algorithm:
>     * **Self generated chain-of-thought improvement suggestions:** In Liu et. al, they do not instruct the VLM to produce chain-of-thought improvements. In fact, in their official implementation, they specifically prompt the VLM to “Respond only with the revised text prompt and exclude any additional commentary”.
>     * **Judge model:** Liu et. al does not incorporate an external judge model to provide signals for the iterative improvements.
>     * **Re-evaluation:** Because of the lack of a judge model, Liu et. al is unable to perform re-evaluation. Both the judge model and re-evaluation have been proven crucial for our algorithm in the ablation study we conduct in Section D in the appendix.
>     * **Parallel search:** Because of the lack of re-evaluation, Liu et. al is unable to perform parallel search since there is no way for them to identify the best prompts from the search. In fact, they can only assume that VLM can monotonically improve the prompt throughout the iterations, which we have proven to be not true in our ablation study. In the case of image classification, which is the main focus of their paper, they have described an alternative way to perform this search by leveraging the validation set and the classification error. However, with the image generation task, they were not able to find a straightforward way to incorporate these designs. As a matter of fact, they use n_restart = 1, n_reset = 1, m = 1 in their official implementation, which effectively makes this algorithm into re-prompting the VLM for several iterations in a single stream, without parallel search or beam search.
>
> To demonstrate the importance of these algorithmic differences, we have tested Liu et. al in our image inversion task with GPT-4V and SDXL-Turbo. To ensure fair comparison, we have also included PRISM with the same iteration budget without parallel search (N = 1, K = 5) and the GPT-4V zero-shot results. Below is the quantitative comparison between our method and Liu et. al. We have included the same results as well as qualitative comparisons in Appendix Section F.
>
> |       Method      |  NLL  | CLIP-I |
> |:-----------------:|:-----:|:------:|
> | Liu et. al (2024) | 2.520 |  0.720 |
> |  GPT-4V Zero Shot | **2.356** |  0.756 |
> |  PRISM (N=1, K=5) | 2.809 |  0.770 |
> |       PRISM (N=6, K=5)       | 2.762 |  **0.776** |
>
> As we can observe, not only does Liu et. al underperform both PRISM versions, it even underperforms GPT-4V zero-shot with the system prompts we design for PRISM. This experiment shows the effectiveness of all components in our algorithm that Liu et. al misses.
>
> We are more than happy to conduct the full image inversion comparison with Liu et. al and update the comparison table in the main text because we think this can significantly strengthen our paper. However, since the full comparison involves slower T2I models such as SD 2.1, we may not have enough time to finish all the experiments before the discussion period ends. We will certainly post an update here if they are finished in time, and will definitely include them in the camera ready version if our paper is accepted. Regardless, the experiment above is sufficient to show the superiority of our algorithm design when compared to Liu et. al in the image inversion task.
>
> We hope this response can answer your concern and please let us know if you have any further questions.

---

> ### Author Response · Authors · 2024-11-24
> **Update on the comparison with Liu et. al (2024)**
>
> We are pleased to share that we have successfully parallelized and accelerated the image generation, and were able to complete the full comparison between our method and Liu et al. (2024). We have updated our manuscript to include this experiment in our main text and we also report the metrics below.
>
> |       Method      |    NLL    |   SD 2.1 CLIP-I   | SDXL-Turbo CLIP-I | Dall-E 2 CLIP-I | Dall-E 2 Failed | Dall-E 3 CLIP-I | Dall-E 3 Failed |
> |:-----------------:|:-----:|---------------|:----------:|:---------------:|:---------------:|:---------------:|:---------------:|
> | Liu et. al (2024) |   **2.520**   |     0.713     |    0.720   |      0.689      |        **0%**       |      0.732      |        **0%**       |
> |       PRISM       |   2.762.  |     **0.749**     |    **0.776**   |      **0.741**      |        2%       |      **0.767**      |        **0%**       |
>
> The results show that PRISM significantly outperforms Liu et al. in terms of CLIP-I scores across all tested T2I models, with only negligible differences in NLL and DALL-E 2 failure rate, where Liu et al. has a slight edge. We sincerely thank the reviewer for suggesting this comparison, as it has strengthened our paper and positioned it better within the LLM-related literature. We hope this experiment addresses the reviewer’s concerns and can convince the reviewer of the unique contributions PRISM makes to the field.

---

> > ### Author Response · Authors · 2024-11-27
> > **Looking forward to your response**
> >
> > Thank you for your time and thoughtful feedback on our paper. One of the reviewers (Reviewer uivB) has expressed interest in hearing your perspectives before finalizing their evaluation. If you have additional thoughts, questions, or concerns, we would greatly appreciate your input to further clarify or improve our work, especially as today is the final day we are able to make changes to the paper PDF. We look forward to your response and are happy to address any additional points that can assist in your review process.

---

> ### Comment · Reviewer_7uVm · 2024-12-01
>
> Thanks for reporting this extra comparison. Re the lack of CoT reasoning in Liu et al, I double checked their paper and implementation, and they have a mismatch between what they report in the paper (Fig 3, 4, and Section 10) that shows CoT and what they do in the implementation (no CoT reasoning). This mismatch is their fault, and therefore I consider your comparison fair and acknowledge your superior performance. Though, I still have some doubts about the magnitude of the contribution of your work compared to them and other related works. The framework is not novel and what gives you best performance at the end of the day are better meta/system-prompts for GPT-4V (since your external judge is the same GPT-4V that does automatic prompt engineering but prompted differently), together with bootstrapping/re-evaluating different initial prompts.
>
> All in all, the method obtains sota results in the task of personalization and prompt inversion, however there is very limited novelty in the proposed approach, which to me outweights the performance improvements. It is painful given the great effort put in the rebuttal and the well-executed paper, but I decided to keep my rating. In case of rejection, I suggest the authors to target a venue where novelty is less important, e.g., TMLR.

---

> ### Author Response · Authors · 2024-12-01
> **Thank you for your response**
>
> Thank you for your thoughtful response and for taking the time to revisit and thoroughly evaluate our work. While we are sorry that we could not fully convince the reviewer of the novelty of our approach, we respectfully disagree with their assessment. We believe that inference-time search algorithm design is an important and non-trivial aspect of LLM-related research. This research direction has demonstrated its potential in fields like reasoning and mathematical problem solving [1,2] but remains under-explored in many other areas. T2I personalization and image inversion is one of these under-explored domains that has significant practical and immediate applications to enhance creativity and accessibility for everyday users. Our paper provides systematic evaluations and ablations of different designs, and proposes an effective and portable algorithm that achieves SOTA performance for this problem.
>
> That being said, we sincerely appreciate the reviewer's constructive feedback. This rebuttal discussion has been truly helpful for strengthening our paper and for better contextualization in the related literature. We are very grateful for the opportunity to improve our work and for your time and effort throughout this process.
>
> Thank you again for your thoughtful review.
>
>
> References:
>
> [1] OpenAI. "Learning to Reason with LLMs." https://openai.com/index/learning-to-reason-with-llms/
>
> [2] Qwen Team. "QwQ: Reflect Deeply on the Boundaries of the Unknown." https://qwenlm.github.io/blog/qwq-32b-preview/

---

### Author Response · Authors · 2024-11-22
**Rebuttal response to all reviewers**

We are grateful to the reviewers for their recognition of our work. In particular, the following strengths were highlighted in the reviews:
1. **Clarity and Presentation:** The paper is well-written, clear, and accessible, with a logical structure that is easy to follow (**Reviewers 7uVm, 9NvK**).
2. **Technical Contributions:**
* PRISM is intuitive, straightforward to implement, and training-free (**Reviewers 2eVw, 9NvK, uivB**).
* The model-agnostic design ensures compatibility with black-box T2I, Judge, and Prompt Engineer models (**Reviewer uivB**).
* PRISM generates fully human-readable prompts, unlike previous bag-of-words techniques, offering higher interpretability and transferability (**Reviewers 7uVm, 9NvK**).
* Useful discussions on alternative judge models (e.g., alternatives to CLIP) (**Reviewers 2eVw**) and related works (**Reviewers uivB**).
3. **Experimental Strengths:**
* Comprehensive, well-executed experiments covering diverse use cases like personalized T2I, prompt inversion, and editing (**Reviewers 7uVm, 2eVw, uivB**).
* Extensibility demonstrated across diverse T2I model families and versions (**Reviewer 2eVw**).
* Quantitative evaluations (NLL, CLIP-I, DINO scores) and qualitative insights highlighting PRISM's strengths, such as structural consistency and fine visual details (**Reviewers 2eVw, uivB**).

We also thank all reviewers for their thoughtful feedback and constructive suggestions, which have helped improve our work. In response, we have carefully addressed all comments and made the following revisions to our manuscript:
1. **Clarification of novelty (Reviewer 7uVm):** We have revised the introduction and the related work section to better emphasize the uniqueness and contributions of our work in comparison to other LLM as optimizer literatures.
2. **Better literature review on LLM as optimizers (Reviewer 7uVm & Reviewer 2eVw):** We have included an additional section (Section F) in the appendix to better contextualize our work among other LLM as optimizer papers.
3. **Additional experimental comparison with Idea2Img (Reviewer 7uVm):** We have added new quantitative and qualitative comparisons between our method and Idea2Img.
4. **Further discussions on the effects of different VLM backbones (Reviewer 2eVw):** We have expanded the discussion in Section D and Section E in the appendix to discuss effects brought by certain limitations of different VLMs as the backbones for the judge.
5. **New ablation study on the prompt length (Reviewer 9NvK):** We have added a new ablation study on the prompt length in Section D in the appendix.
6. **New ablation comparison between the judge score distribution of the first iteration and the final prompts (Reviewer uivB):** We have added a new ablation comparison between the judge score distribution of the first iteration and the final prompts in Section D in the appendix.
7. **New latency comparison (Reviewer uivB):** We have added a new latency comparison in Section E in the appendix.
8. **Two new applications, prompt distillation and multi-concept generation (Reviewer 9NvK):** We have added another section to demonstrate two new applications, prompt distillation and multi-concept generation, to Appendix Section C.3.
9. **New latency comparison (Reviewer uivB):** We have added the latency comparison in Section E.
10. **Reorganizing the figures in the Appendix for easier references.**

Each addition is marked red in our updated manuscript and also discussed in detail in our individual responses to the reviewers. We believe these updates address the key concerns and reflect all the suggestions the reviewers have made. We are happy to engage in further discussions to clarify any additional comments or questions.

---

### Meta-Review · Area_Chair_QVXh · 2024-12-17

**Metareview:**

This paper introduces an in-context learning approach to iteratively update prompts for to improve text-to-image generation. The paper was reviewed by three knowledgeable reviewers, who acknowledged that the paper was well presented and clear (2eVw, 9NvK), and the method's design  simple (9NvK) and model agnostic (uivB). The main concerns raised by the reviewers were:
1. Unclear novelty of the proposed approach (7uVm)
2. Positioning w.r.t. prior art
3. Limited discussion of weaknesses (2eVw)
4. Unconvincing experimental evidence: marginal improvements (7uVm), missing ablations (9NvK) or resource-fixed comparisons with CLIP-Interrogator (uivB)

During rebuttal/discussion phase, the authors partially addressed the reviewers' concerns by arguing for the novelty of the proposed approach (targeting a different task than prior work) and their achieved improvements, adding comparisons with some existing methods pointed by the reviewers, analyzing the effect of iterations, adding ablations on the effect of prompt length, clarifying questions related to the method/experiments, and discussing resource and latency comparisons. After discussion, the reviewers remain unconvinced. In particular, concerns w.r.t. novelty persist (LLM as optimizers have been applied for T2I models, only the end task differs), and the experimental results, which should include more comparisons with prior work as well as CLIP Interrogator (the latter would need to be evaluated by adjusting the captioning model). For these reasons, the reviewers lean towards rejection during the discussion period. The AC agrees with the reviewers' assessment and recommends to reject. The AC encourages the authors to consider the feedback received to improve future iterations of their work.

**Additional Comments On Reviewer Discussion:**

See above.

---

### Decision · Program_Chairs · 2025-01-22

Reject